# Convergence Analysis of Two-layer Neural Networks with ReLU Activation

**Yuanzhi Li**
Computer Science Department
Princeton University
yuanzhil@cs.princeton.edu

**Yang Yuan**
Computer Science Department
Cornell University
yangyuan@cs.cornell.edu

## Abstract

In recent years, stochastic gradient descent (SGD) based techniques has become the standard tools for training neural networks. However, formal theoretical understanding of why SGD can train neural networks in practice is largely missing.

In this paper, we make progress on understanding this mystery by providing a convergence analysis for SGD on a rich subset of two-layer feedforward networks with ReLU activations. This subset is characterized by a special structure called "identity mapping". We prove that, if input follows from Gaussian distribution, with standard $O(1/\sqrt{d})$ initialization of the weights, SGD converges to the global minimum in polynomial number of steps. Unlike normal vanilla networks, the "identity mapping" makes our network asymmetric and thus the global minimum is unique. To complement our theory, we are also able to show experimentally that multi-layer networks with this mapping have better performance compared with normal vanilla networks.

Our convergence theorem differs from traditional non-convex optimization techniques. We show that SGD converges to optimal in "two phases": In phase I, the gradient points to the wrong direction, however, a potential function $g$ gradually decreases. Then in phase II, SGD enters a nice one point convex region and converges. We also show that the identity mapping is necessary for convergence, as it moves the initial point to a better place for optimization. Experiment verifies our claims.

## 1 Introduction

Deep learning is the mainstream technique for many machine learning tasks, including image recognition, machine translation, speech recognition, etc. [17]. Despite its success, the theoretical understanding on how it works remains poor. It is well known that neural networks have great expressive power [22, 7, 3, 8, 31]. That is, for every function there exists a set of weights on the neural network such that it approximates the function everywhere. However, it is unclear how to obtain the desired weights. In practice, the most commonly used method is stochastic gradient descent based methods (e.g., SGD, Momentum [40], Adagrad [10], Adam [25]), but to the best of our knowledge, there were no theoretical guarantees that such methods will find good weights.

In this paper, we give the first convergence analysis of SGD for two-layer feedforward network with ReLU activations. For this basic network, it is known that even in the simplified setting where the weights are initialized symmetrically and the ground truth forms orthonormal basis, gradient descent might get stuck at saddle points [41].

Inspired by the structure of residual network (ResNet) [21], we add an extra identity mapping for the hidden layer (see Figure 1). Surprisingly, we show that simply by adding this mapping, with the standard initialization scheme and small step size, SGD always converges to the ground truth. In other

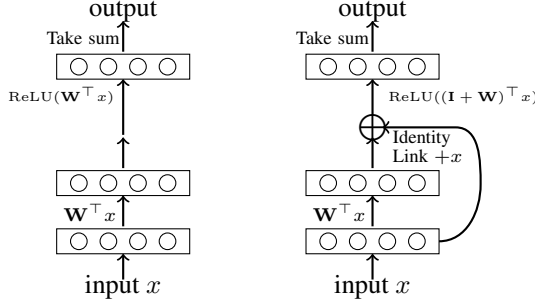
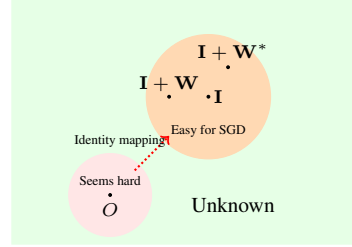

Figure 1: Vanilla network (left), with identity mapping (right)    Figure 2: Illustration for our result.

words, the optimization becomes significantly easier, after adding the identity mapping. See Figure 2, based on our analysis, the region near the identity matrix $\mathbf{I}$ contains only one global minimum without any saddle points or local minima, thus is easy for SGD to optimize. The role of the identity mapping here, is to move the initial point to this easier region (better initialization).

Other than being feedforward and shallow, our network is different from ResNet in the sense that our identity mapping skips one layer instead of two. However, as we will show in Section 5.1, the skip-one-layer identity mapping already brings significant improvement to vanilla networks.

Formally, we consider the following function.

$$f(x, \mathbf{W}) = \|\text{ReLU}((\mathbf{I} + \mathbf{W})^\top x)\|_1 \tag{1}$$

where $\text{ReLU}(v) = \max(v, 0)$ is the ReLU activation function. $x \in \mathbb{R}^d$ is the input vector sampled from a Gaussian distribution, and $\mathbf{W} \in \mathbb{R}^{d \times d}$ is the weight matrix, where $d$ is the number of input units. Notice that $\mathbf{I}$ adds $e_i$ to column $i$ of $\mathbf{W}$, which makes $f$ *asymmetric* in the sense that by switching any two columns in $\mathbf{W}$, we get different functions.

Following the standard setting [34, 41], we assume that there exists a two-layer teacher network with weight $\mathbf{W}^*$. We train the student network using $\ell_2$ loss:

$$\mathsf{L}(\mathbf{W}) = \mathbb{E}_x[(f(x, \mathbf{W}) - f(x, \mathbf{W}^*))^2] \tag{2}$$

We will define a potential function $g$, and show that if $g$ is small, the gradient points to partially correct direction and we get closer to $\mathbf{W}^*$ after every SGD step. However, $g$ could be large and thus gradient might point to the reverse direction. Fortunately, we also show that if $g$ is large, by doing SGD, it will keep decreasing until it is small enough while maintaining the weight $\mathbf{W}$ in a nice region. We call the process of decreasing $g$ as Phase I, and the process of approaching $\mathbf{W}^*$ as Phase II. See Figure 3 and simulations in Section 5.3.

Our two phases framework is fundamentally different from any type of local convergence, as in Phase I, the gradient is pointing to the wrong direction to $\mathbf{W}^*$, so the path from $\mathbf{W}$ to $\mathbf{W}^*$ is non-convex, and SGD takes a long detour to arrive $\mathbf{W}^*$. This framework could be potentially useful for analyzing other non-convex problems.

To support our theory, we have done a few other experiments and got interesting observations. For example, as predicted by our theorem, we found that for multilayer feedforward network with identity mappings, zero initialization performs as good as random initialization. At the first glance, it contradicts the common belief "random initialization is necessary to break symmetry", but actually the identity mapping itself serves as the asymmetric component. See Section 5.4.

Another common belief is that neural network has lots of local minima and saddle points [9], so even if there exists a global minimum, we may not be able to arrive there. As a result, even when the teacher network is shallow, the student network usually needs to be deeper, otherwise it will underfit. However, both our theorem and our experiment show that if the shallow teacher network is in a pretty large region near identity (Figure 2), SGD always converges to the global minimum by initializing the weights $\mathbf{I} + \mathbf{W}$ in this region, with equally shallow student network. By contrast, wrong initialization gets stuck at local minimum and underfit. See Section 5.2.

**Related Work**

**Expressivity**. Even two-layer network has great expressive power. For example, two-layer network with sigmoid activations could approximate any continuous function [22, 7, 3]. ReLU is the state-of-the-art activation function [30, 13], and has great expressive power as well [29, 32, 31, 4, 26].

**Learning**. Most previous results on learning neural network are negative [39, 28, 38], or positive but with algorithms other than SGD [23, 43, 37, 14, 15, 16], or with strong assumptions on the model [1, 2]. [35] proved that with high probability, there exists a continuous decreasing path from random initial point to the global minimum, but SGD may not follow this path. Recently, Zhong et al. showed that with initialization point found using tensor decomposition, gradient descent could find the ground truth for one hidden layer network [44].

**Linear network and independent activation**. Some previous works simplified the model by ignoring the activation functions and considering deep linear networks [36, 24] or deep linear residual networks [19], which can only learn linear functions. Some previous results are based on independent activation assumption that the activations of ReLU and the input are independent [5, 24].

**Saddle points**. It is observed that saddle point is not a big problem for neural networks [9, 18]. In general, if the objective is strict-saddle [11], SGD could escape all saddle points.

## 2 Preliminaries

Denote $x$ as the input vector in $\mathbb{R}^d$. For now, we first consider $x$ sampled from normal distribution $\mathcal{N}(0, \mathbf{I})$. Denote $\mathbf{W}^* = (w_1^*, \cdots, w_n^*) \in \mathbb{R}^{d \times d}$ as the weights for the teacher network, $\mathbf{W} = (w_1, \cdots, w_n) \in \mathbb{R}^{d \times d}$ as the weights for the student network, where $w_i^*, w_i \in \mathbb{R}^d$ are column vectors. $f(x, \mathbf{W}^*), f(x, \mathbf{W})$ are defined in (1), representing the teacher and student network.

We want to know whether a randomly initialized $\mathbf{W}$ will converge to $\mathbf{W}^*$, if we run SGD with $l_2$ loss defined in (2). Alternatively, we can write the loss $\mathsf{L}(\mathbf{W})$ as

$$\mathbb{E}_x[(\Sigma_i \mathrm{ReLU}(\langle e_i + w_i, x \rangle) - \Sigma_i \mathrm{ReLU}(\langle e_i + w_i^*, x \rangle))^2]$$

Taking derivative with respect to $w_j$, we get

$$\nabla \mathsf{L}(\mathbf{W})_j = 2\mathbb{E}_x \left[ \left( \sum_i \mathrm{ReLU}(\langle e_i + w_i, x \rangle) - \sum_i \mathrm{ReLU}(\langle e_i + w_i^*, x \rangle) \right) x \mathbb{1}_{\langle e_j + w_j, x \rangle \geq 0} \right]$$

where $\mathbb{1}_e$ is the indicator function that equals $1$ if the event $e$ is true, and $0$ otherwise. Here $\nabla \mathsf{L}(\mathbf{W}) \in \mathbb{R}^{d \times d}$, and $\nabla \mathsf{L}(\mathbf{W})_j$ is its $j$-th column.

Denote $\theta_{i,j}$ as the angle between $e_i + w_i$ and $e_j + w_j$, $\theta_{i^*,j}$ as the angle between $e_i + w_i^*$ and $e_j + w_j$. Denote $\bar{v} = \frac{v}{\|v\|_2}$. Denote $\overline{\mathbf{I} + \mathbf{W}^*}$ and $\overline{\mathbf{I} + \mathbf{W}^*}$ as the column-normalized version of $\mathbf{I} + \mathbf{W}^*$ and $\mathbf{I} + \mathbf{W}$ such that every column has unit norm. Since the input is from a normal distribution, one can compute the expectation inside the gradient as follows.

**Lemma 2.1** (Eqn (13) from [41]). *If $x \sim \mathcal{N}(0, \mathbf{I})$, then* $-\nabla \mathsf{L}(\mathbf{W})_j = \sum_{i=1}^d \left( \frac{\pi}{2}(w_i^* - w_i) + \left( \frac{\pi}{2} - \theta_{i^*,j} \right)(e_i + w_i^*) - \left( \frac{\pi}{2} - \theta_{i,j} \right)(e_i + w_i) + \left( \|e_i + w_i^*\|_2 \sin \theta_{i^*,j} - \|e_i + w_i\|_2 \sin \theta_{i,j} \right) \overline{e_j + w_j} \right)$

**Remark.** Although the gradient of ReLU is not well defined at the point of zero, if we assume input $x$ is from the Gaussian distribution, the loss function becomes smooth, and the gradient is well defined everywhere.

Denote $u \in \mathbb{R}^d$ as the all one vector. Denote $\mathrm{Diag}(\mathbf{W})$ as the diagonal matrix of matrix $\mathbf{W}$, $\mathrm{Diag}(v)$ as a diagonal matrix whose main diagonal equals to the vector $v$. Denote $\mathrm{Off\text{-}Diag}(\mathbf{W}) \triangleq \mathbf{W} - \mathrm{Diag}(\mathbf{W})$. Denote $[d]$ as the set $\{1, \cdots, d\}$. Throughout the paper, we abuse the notation of inner product between matrices $\mathbf{W}, \mathbf{W}^*, \nabla \mathsf{L}(\mathbf{W})$, such that $\langle \nabla \mathsf{L}(\mathbf{W}), \mathbf{W} \rangle$ means the summation of the entrywise products. $\|\mathbf{W}\|_2$ is the spectral norm of $\mathbf{W}$, and $\|\mathbf{W}\|_F$ is the Frobenius norm of $\mathbf{W}$. We define the potential function $g$ and variables $g_j, \mathbf{A}_j, \mathbf{A}$ below, which will be useful in the proof.

**Definition 2.2.** *We define the potential function* $g \triangleq \sum_{i=1}^d (\|e_i + w_i^*\|_2 - \|e_i + w_i\|_2)$, *and variable* $g_j \triangleq \sum_{i \neq j} (\|e_i + w_i^*\|_2 - \|e_i + w_i\|_2)$.

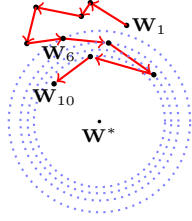
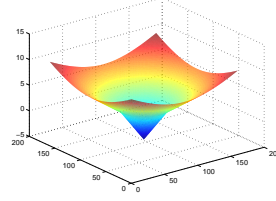

Figure 3: Phase I: $\mathbf{W}_1 \to \mathbf{W}_6$, $\mathbf{W}$ may go to the wrong direction but the potential is shrinking. Phase II: $\mathbf{W}_6 \to \mathbf{W}_{10}$, $\mathbf{W}$ gets closer to $\mathbf{W}^*$ in every step by one point convexity.

Figure 4: The function is one point strongly convex as every point's negative gradient points to the center, but not convex as any line between the center and the red region is below surface.

**Definition 2.3.** *Denote* $\mathbf{A}_j \triangleq \sum_{i \neq j} ((e_i + w_i^*)\overline{e_i + w_i^*}^\top - (e_i + w_i)\overline{e_i + w_i}^\top), \mathbf{A} \triangleq \sum_{i=1}^{d} ((e_i + w_i^*)\overline{e_i + w_i^*}^\top - (e_i + w_i)\overline{e_i + w_i}^\top) = (\mathbf{I} + \mathbf{W}^*)\overline{\mathbf{I} + \mathbf{W}^*}^\top - (\mathbf{I} + \mathbf{W})\overline{\mathbf{I} + \mathbf{W}}^\top$.

In this paper, we consider the standard SGD with mini batch method for training the neural network. Assume $\mathbf{W}_0$ is the initial point, and in step $t > 0$, we have the following updating rule:

$$\mathbf{W}_{t+1} = \mathbf{W}_t - \eta_t \mathbf{G}_t$$

where the stochastic gradient $\mathbf{G}_t = \nabla \mathsf{L}(\mathbf{W}_t) + \mathbf{E}_t$ with $\mathbb{E}[\mathbf{E}_t] = \mathbf{0}$ and $\|\mathbf{E}_t\|_F \leq \varepsilon$. Let $G_2 \triangleq 6d\gamma + \varepsilon, G_F \triangleq 6d^{1.5}\gamma + \varepsilon$, where $\gamma$ is the upper bound of $\|\mathbf{W}^*\|_2$ and $\|\mathbf{W}_0\|_2$ (defined later). As we will see in Lemma C.2, they are the upper bound of $\|\mathbf{G}_t\|_2$ and $\|\mathbf{G}_t\|_F$ respectively.

It's clear that $\mathsf{L}$ is not convex, In order to get convergence guarantees, we need a weaker condition called *one point convexity*.

**Definition 2.4** (One point strongly convexity). *A function $f(x)$ is called $\delta$-one point strongly convex in domain $\mathbf{D}$ with respect to point $x^*$, if $\forall x \in \mathbf{D}, \langle -\nabla f(x), x^* - x \rangle > \delta \|x^* - x\|_2^2$.*

By definition, if a function $f$ is strongly convex, it is also one point strongly convex in the entire space with respect to the global minimum. However, the reverse is not necessarily true, e.g., see Figure 4. If a function is one point strongly convex, then in every step a positive fraction of the negative gradient is pointing to the optimal point. As long as the step size is small enough, we will finally arrive the optimal point, possibly by a winding path. See Figure 3 for illustration, where starting from $\mathbf{W}_6$ (Phase II), we get closer to $\mathbf{W}^*$ in every step. Formally, we have the following lemma.

**Lemma 2.5.** *For function $f(\mathbf{W})$, consider the SGD update $\mathbf{W}_{t+1} = \mathbf{W}_t - \eta \mathbf{G}_t$, where $\mathbb{E}[\mathbf{G}_t] = \nabla f(\mathbf{W}_t)$, $\mathbb{E}[\|\mathbf{G}_t\|_F^2] \leq G^2$. Suppose for all $t$, $\mathbf{W}_t$ is always inside the $\delta$-one point strongly convex region with diameter $D$, i.e., $\|\mathbf{W}_t - \mathbf{W}^*\|_F \leq D$. Then for any $\alpha > 0$ and any $T$ such that $T^\alpha \log T \geq \frac{D^2 \delta^2}{(1+\alpha)G^2}$, if $\eta = \frac{(1+\alpha)\log T}{\delta T}$, we have $\mathbb{E}\|\mathbf{W}_T - \mathbf{W}^*\|_F^2 \leq \frac{(1+\alpha)\log T G^2}{\delta^2 T}$.*

The proof can be found in Appendix J. Lemma 2.5 uses fixed step size, so it easily fits the standard practical scheme that shrinks $\eta$ by a factor of 10 after every a few epochs. For example, we may apply Lemma 2.5 every time $\eta$ gets changed. Notice that our lemma does not imply that $\mathbf{W}_T$ will converge to $\mathbf{W}^*$. Instead, it only says $\mathbf{W}_T$ will be sufficiently close to $\mathbf{W}^*$ with small step size $\eta$.

## 3 Main Theorem

**Theorem 3.1** (Main Theorem). *There exists constants $\gamma > \gamma_0 > 0$ such that If $x \sim \mathcal{N}(0, \mathbf{I})$, $\|\mathbf{W}_0\|_2, \|\mathbf{W}^*\|_2 \leq \gamma_0$, $d \geq 100$, $\varepsilon \leq \gamma^2$, then SGD for $\mathsf{L}(\mathbf{W})$ will find the ground truth $\mathbf{W}^*$ by two phases. In Phase I, by setting $\eta \leq \frac{\gamma^2}{G_2^2}$, the potential function will keep decreasing until it is smaller than $197\gamma^2$, which takes at most $\frac{1}{16\eta}$ steps. In Phase II, for any $\alpha > 0$ and any $T$ such that $T^\alpha \log T \geq \frac{36d}{100^4(1+\alpha)G_F^2}$, if we set $\eta = \frac{(1+\alpha)\log T}{\delta T}$, we have $\mathbb{E}\|\mathbf{W}_T - \mathbf{W}^*\|_F^2 \leq \frac{100^2(1+\alpha)\log T G_F^2}{9T}$.*

**Remarks.** Randomly initializing the weights with $O(1/\sqrt{d})$ is standard in deep learning, see [27, 12, 20]. It is also well known that if the entries are initialized with $O(1/\sqrt{d})$, the spectral norm

of the random matrix is $O(1)$ [33]. So our result matches with the common practice. Moreover, as we will show in Section 5.5, networks with small average spectral norm already have good performance. Thus, our assumption $\|\mathbf{W}^*\|_2 = O(1)$ is reasonable. Notice that here we assume the *spectral norm* of $\mathbf{W}^*$ to be constant, which means the Frobenius norm $\|\mathbf{W}^*\|_F$ could be as big as $O(\sqrt{d})$.

The assumption that the input follows a Gaussian distribution is not necessarily true in practice (Although this is a common assumption appeared in the previous papers [5, 41, 42], and also considered plausible in [6]). We could easily generalize the analysis to rotation invariant distributions, and potentially more general distributions (see Section 6). Moreover, previous analyses either ignore the nonlinear activations and thus consider linear model [36, 24, 19], or directly [5, 24] or indirectly [41][1] assume that the activations are independent. By contrast, in our model the ReLU activations are highly correlated[2] as $\|\mathbf{W}\|_2, \|\mathbf{W}^*\|_2 = \Omega(1)$. As pointed out by [6], eliminating the unrealistic assumptions on activation independence is the central problem of analyzing the loss surface of neural network, which was not fully addressed by the previous analyses.

To prove the main theorem, we split the process and present the following two theorems, which will be proved in Appendix C and D.

**Theorem 3.2** (Phase I). *There exists a constant $\gamma > \gamma_0 > 0$ such that If $\|\mathbf{W}_0\|_2, \|\mathbf{W}^*\|_2 \leq \gamma_0$, $d \geq 100$, $\eta \leq \frac{\gamma^2}{G_2^2}$, $\varepsilon \leq \gamma^2$, then $g_t$ will keep decreasing by a factor of $1 - 0.5\eta d$ for every step, until $g_{t_1} \leq 197\gamma^2$ for step $t_1 \leq \frac{1}{16\eta}$. After that, Phase II starts. That is, for every $T > t_1$, we have $\|\mathbf{W}_T\|_2 \leq \frac{1}{100}$ and $g_T \leq 0.1$.*

**Theorem 3.3** (Phase II). *There exists a constant $\gamma$ such that if $\|\mathbf{W}\|_2, \|\mathbf{W}^*\|_2 \leq \gamma$, and $g \leq 0.1$, then $\langle -\nabla \mathsf{L}(\mathbf{W}), \mathbf{W}^* - \mathbf{W} \rangle = \sum_{j=1}^d \langle -\nabla \mathsf{L}(\mathbf{W})_j, w_j^* - w_j \rangle > 0.03\|\mathbf{W}^* - \mathbf{W}\|_F^2$.*

With these two theorems, we get the main theorem immediately.

*Proof for Theorem 3.1.* By Theorem 3.2, we know the statement for Phase I is true, and we will enter phase II in $\frac{1}{16\eta}$ steps. After entering Phase II, based on Theorem 3.3, we simply use Lemma 2.5 by setting $\delta = 0.03$, $D = \frac{\sqrt{d}}{50}$, $G = G_F$ to get the convergence guarantee. $\qquad\square$

# 4 Overview of the Proofs

**General Picture**. In many convergence analyses for non-convex functions, one would like to show that $\mathsf{L}$ is one point strongly convex, and directly apply Lemma 2.5 to get the convergence result. However, this is not true for 2-layer neural network, as the gradient may point to the wrong direction, see Section 5.3.

So when is our $\mathsf{L}$ one point convex? Consider the following thought experiment: First, suppose $\|\mathbf{W}\|_2, \|\mathbf{W}^*\|_2 \to 0$, we know $\|w_i\|_2, \|w_i^*\|_2$ also go to 0. Thus, $e_i + w_i$ and $e_i + w_i^*$ are close to $e_i$. As a result, $\theta_{i,j}, \theta_{i^*,j} \approx \frac{\pi}{2}$, and $\theta_{i^*,i} \approx 0$. Based on Lemma 2.1, this gives us a naïve approximation of the negative gradient, i.e., $-\nabla \mathsf{L}(\mathbf{W})_j \approx \frac{\pi}{2}(w_j^* - w_j) + \frac{\pi}{2}\sum_{i=1}^d (w_i^* - w_i) + \overline{e_j + w_j}\sum_{i \neq j}(\|e_i + w_i^*\|_2 - \|e_i + w_i\|_2)$ .

While the first two terms $\frac{\pi}{2}(w_j^* - w_j)$ and $\frac{\pi}{2}\sum_{i=1}^d (w_i^* - w_i)$ have positive inner product with $\mathbf{W}^* - \mathbf{W}$, the last term $g_j = \overline{e_j + w_j}\sum_{i \neq j}(\|e_i + w_i^*\|_2 - \|e_i + w_i\|_2)$ can point to arbitrary direction. If the last term is small, it can be covered by the first two terms, and $\mathsf{L}$ becomes one point strongly convex. So we define a potential function closely related to the last term: $g = \sum_{i=1}^d (\|e_i + w_i^*\|_2 - \|e_i + w_i\|_2)$. We show that if $g$ is small enough, $\mathsf{L}$ is also one point strongly convex (Theorem 3.3).

However, from random initialization, $g$ can be as large as of $\Omega(\sqrt{d})$, which is too big to be covered. Fortunately, we show that if $g$ is big, it will gradually decrease simply by doing SGD on $\mathsf{L}$. More specifically, we introduce a *two phases* convergence analysis framework:

Figure 5: Lower bounds of inner product using Taylor expansion

1. In Phase I, the potential function $g$ is decreasing to a small value.
2. In Phase II, $g$ remains small, so L is one point convex and thus $\mathbf{W}$ starts to converge to $\mathbf{W}^*$.

We believe that this framework could be helpful for other non-convex problems.

**Technical difficulty: Phase I**. Our key technical challenge is to show that in Phase I, the potential function actually decreases to $O(1)$ after polynomial number of iterations. However, we cannot show this by merely looking at $g$ itself. Instead, we introduce an auxiliary variable $s = (\mathbf{W}^* - \mathbf{W})u$, where $u$ is the all one vector. By doing a careful calculation, we get their joint update rules (Lemma C.3 and Lemma C.4):

$$\begin{cases} s_{t+1} & \approx s_t - \frac{\pi \eta d}{2}s_t + \eta O(\sqrt{d}g_t + \sqrt{d}\gamma) \\ g_{t+1} & \approx g_t - \eta d g_t + \eta O(\gamma\sqrt{d}\|s_t\|_2 + d\gamma^2) \end{cases}$$

Solving this dynamics, we can show that $g_t$ will approach to (and stay around) $O(\gamma)$, thus we enter Phase II.

**Technical difficulty: Phase II**. Although the overall approximation in the thought experiment looks simple, the argument is based on an over simplified assumption that $\theta_{i^*,j}, \theta_{i,j} \approx \frac{\pi}{2}$ for $i \neq j$. However, when $\mathbf{W}^*$ has constant spectral norm, even when $\mathbf{W}$ is very close to $\mathbf{W}^*$, $\theta_{i,j^*}$ could be constantly far away from $\frac{\pi}{2}$, which prevents us from applying this approximation directly. To get a formal proof, we use the standard Taylor expansion and control the higher order terms. Specifically, we write $\theta_{i^*,j}$ as $\theta_{i^*,j} = \arccos\langle \overline{e_i + w_i^*}, \overline{e_j + w_j}\rangle$ and expand $\arccos$ at point 0, thus,

$$\theta_{i^*,j} = \frac{\pi}{2} - \langle \overline{e_i + w_i^*}, \overline{e_j + w_j}\rangle + O(\langle \overline{e_i + w_i^*}, \overline{e_j + w_j}\rangle^3)$$

However, even when $\mathbf{W} \approx \mathbf{W}^*$, the higher order term $O(\langle \overline{e_i + w_i^*}, \overline{e_j + w_j}\rangle^3)$ still can be as large as a constant, which is too big for us. Our trick here is to consider the "joint Taylor expansion":

$$\theta_{i^*,j} - \theta_{i,j} = \langle \overline{e_i + w_i} - \overline{e_i + w_i^*}, \overline{e_j + w_j}\rangle + O(|\langle \overline{e_i + w_i^*}, \overline{e_j + w_j}\rangle^3 - \langle \overline{e_i + w_i}, \overline{e_j + w_j}\rangle^3|)$$

As $\mathbf{W}$ approaches $\mathbf{W}^*$, $|\langle \overline{e_i + w_i^*}, \overline{e_j + w_j}\rangle^3 - \langle \overline{e_i + w_i}, \overline{e_j + w_j}\rangle^3|$ also tends to zero, therefore our approximation has bounded error.

In the thought experiment, we already know that the constant part in the Taylor expansion of $\nabla \mathsf{L}(\mathbf{W})$ is $\frac{\pi}{2} - O(g)$-one point convex. We show that after taking inner product with $\mathbf{W}^* - \mathbf{W}$, the first order terms are lower bounded by (roughly) $-1.3\|\mathbf{W}^* - \mathbf{W}\|_F^2$ and the higher order terms are lower bounded by $-0.085\|\mathbf{W}^* - \mathbf{W}\|_F^2$. Adding them together, we can see that $\mathsf{L}(\mathbf{W})$ is one point convex as long as $g$ is small. See Figure 5.

**Geometric Lemma**. In order to get through the whole analysis, we need tight bounds on a few common terms that appear everywhere. Instead of using naïve algebraic techniques, we come up with a nice geometric proof to get nearly optimal bounds. Due to space limit, we defer it to Appendix E.

## 5 Experiments

In this section, we present several simulation results to support our theory. Our code can be found in the supplementary materials.

### 5.1 Importance of identity mapping

In this experiment, we compare the standard ResNet [21] and *single skip model* where identity mapping skips only one layer. See Figure 6 for the single skip model. We also ran the vanilla network, where the identity mappings are completely removed.

Table 1: Test error of three 56-layer networks on Cifar-10

|  | ResNet | Single skip | Vanilla |
|---|---|---|---|
| Test Err | 6.97% | 9.01% | 12.04% |

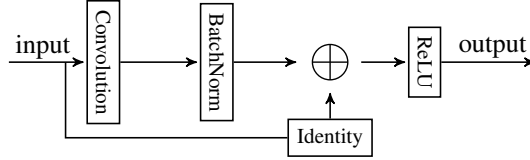

Figure 6: Illustration of one block in single skip model in Sec 5.1

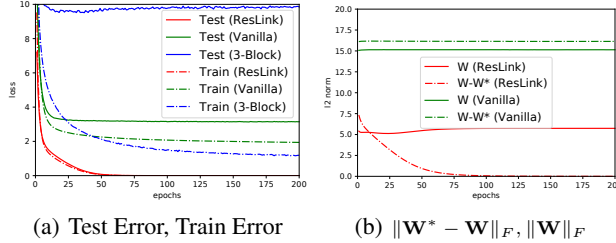

(a) Test Error, Train Error       (b) $\|\mathbf{W}^* - \mathbf{W}\|_F, \|\mathbf{W}\|_F$

Figure 7: Verifying the global convergence

In this experiment, we choose Cifar-10 as the dataset, and all the networks have 56-layers. Other than the identity mappings, all other settings are identical and default. We run the experiments for 5 times and report the average test error. As we can see in Table 1, compared with vanilla network, by simply using a single skip identity mapping, one can already improve the test error by 3.03%, and is 2.04% close to the ResNet. So single skip identity mapping brings significant improvement on test accuracy.

## 5.2 Global minimum convergence

In this experiment, we verify our main theorem that for two-layer teacher network and student network with identity mappings, as long as $\|\mathbf{W}_0\|_2, \|\mathbf{W}^*\|_2$ is small, SGD always converges to the global minimum $\mathbf{W}^*$, thus gives almost 0 training error and test error. We consider three student networks. The first one (ResLink) is defined using (2), the second one (Vanilla) is the same model without the identity mapping. The last one (3-Block) is a three block network with each block containing a linear layer (500 hidden nodes), a batch normalization and a ReLU layer. The teacher network always shares the same structure as the student network.

The input dimension is 100. We generated a fixed $\mathbf{W}^*$ for all the trials with $\|\mathbf{W}^*\|_2 \approx 0.6, \|\mathbf{W}^*\|_F \approx 5.7$. We generated a training set of size $100,000$, and test set of size $10,000$, sampled from a Gaussian distribution. We use batch size 200, step size 0.001. We run ResLink for 5 times with random initialization ($\|\mathbf{W}\|_2 \approx 0.6$ and $\|\mathbf{W}\|_F \approx 5$), and plot the curves by taking the average.

Figure 7(a) shows test error and training error of the three networks. Comparing Vanilla with 3-Block, we find that 3-Block is more expressive, so its training error is smaller compared with vanilla network; but it suffers from overfitting and has bigger test error. This is the standard overfitting vs underfitting tradeoff. Surprisingly, with only one hidden layer, ResLink has both zero test error and training error. If we look at Figure 7(b), we know the distance between $\mathbf{W}$ and $\mathbf{W}^*$ converges to 0, meaning ResLink indeed finds the global optimal in all 5 trials. By contrast, for vanilla network, which is essentially the same network with different initialization, $\|\mathbf{W} - \mathbf{W}^*\|_2$ does not converge to zero[3]. This is exactly what our theory predicted.

## 5.3 Verify the dynamics

In this experiment, we verify our claims on the dynamics. Based on the analysis, we construct a $1500 \times 1500$ matrix $\mathbf{W}$ s.t. $\|\mathbf{W}\|_2 \approx 0.15, \|\mathbf{W}\|_F \approx 5$, and set $\mathbf{W}^* = 0$. By plugging them into (2), one can see that even in this simple case that $\mathbf{W}^* = 0$, initially the gradient is pointing to the wrong direction, i.e., not one point convex. We then run SGD on $\mathbf{W}$ by using samples $x$ from Gaussian distribution, with batch size 300, step size 0.0001.

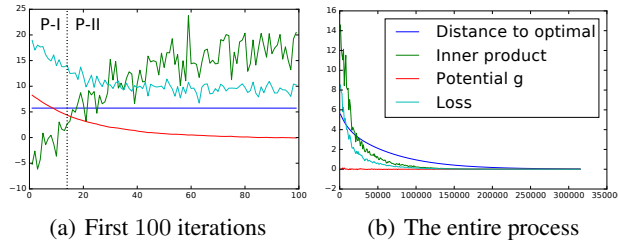

| (a) First 100 iterations | (b) The entire process |

Figure 8: Verifying the dynamics

Figure 8(a) shows the first 100 iterations. We can see that initially the inner product defined in Definition 2.4 is negative, then after about 15 iterations, it turns positive, which means $\mathbf{W}$ is in the one point strongly convex region. At the same time, the potential $g$ keeps decreasing to a small value, while the distance to optimal (which also equals to $\|\mathbf{W}\|_F$ in this experiment) is not affected. They precisely match with our description of Phase I in Theorem 3.2.

After that, we enter Phase II and slowly approach to $\mathbf{W}^*$, see Figure 8(b). Notice that the potential $g$ is always very small, the inner product is always positive, and the distance to optimal is slowly decreasing. Again, they precisely match with our Theorem 3.3.

## 5.4 Zero initialization works

In this experiment, we used a simple 5-block neural network on MNIST, where every block contains a $784 * 784$ feedforward layer, an identity mapping, and a ReLU layer. Cross entropy criterion is used. We compare zero initialization with standard $O(1/\sqrt{d})$ random initialization. We found that for zero initialization, we can get $1.28\%$ test error, while for random initialization, we can get $1.27\%$ test error. Both results were obtained by taking average among 5 runs and use step size $0.1$, batch size $256$. If the identity mapping is removed, zero initialization no longer works.

## 5.5 Spectral norm of $\mathbf{W}^*$

We also applied the exact model $f$ defined in (1) to distinguish two classes in MNIST. For any input image $x$, We say it's in class A if $f(x, \mathbf{W}) < T_{A,B}$, and in class B otherwise. Here $T_{A,B}$ is the optimal threshold for the function $f(x, \mathbf{0})$ to distinguish $A$ and $B$. If $\mathbf{W} = \mathbf{0}$, we get $7\%$ training error for distinguish class $0$ and class $1$. However, it can be improved to $1\%$ with $\|\mathbf{W}\|_2 = 0.6$. We tried this experiment for all possible $45$ pairs of classes in MNIST, and improve the average training error from $34\%$ (using $\mathbf{W} = \mathbf{0}$) to $14\%$ (using $\|\mathbf{W}\|_2 = 0.6$). Therefore our model with $\|\mathbf{W}\|_2 = \Omega(1)$ has reasonable expressive power, and is substantially different from just using the identity mapping alone.

# 6 Discussions

The assumption that the input is Gaussian can be relaxed in several ways. For example, when the distribution is $\mathcal{N}(0, \Sigma)$ where $\|\Sigma - \mathbf{I}\|_2$ is bounded by a small constant, the same result holds with slightly worse constants. Moreover, since the analysis relies Lemma 2.1, which is proved by converting the original input space into polar space, it is easy to generalize the calculation to rotation invariant distributions. Finally, for more general distributions, as long as we could explicitly compute the expectation, which is in the form of $O(\mathbf{W}^* - \mathbf{W})$ plus certain potential function, our analysis framework may also be applied.

There are many exciting open problems. For example, Our paper is the first one that gives solid SGD analysis for neural network with nonlinear activations, without unrealistic assumptions like independent activation assumption. It would be great if one could further extend it to multiple layers, which would be a major breakthrough of understanding optimization for deep learning. Moreover, our two phase framework could be applied to other non-convex problems as well.

## Acknowledgement

The authors want to thank Robert Kleinberg, Kilian Weinberger, Gao Huang, Adam Klivans and Surbhi Goel for helpful discussions, and the anonymous reviewers for their comments.

## Footnotes

[1]They assume input is Gaussian and the $\mathbf{W}^*$ is orthonormal, which means the activations are independent in teacher network.

[2] Let $\sigma_i$ be the output of i-th ReLU unit, then in our setting, $\sum_{i,j}\mathrm{Cov}[\sigma_i, \sigma_j]$ can be as large as $\Omega(d)$, which is far from being independent.

[3]To make comparison meaningful, we set $\mathbf{W} - \mathbf{I}$ to be the actual weight for Vanilla as its identity mapping is missing, which is why it has a much bigger initial norm.

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
