[Supplementary Material]

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

Figure 9: Flowchart of the proofs

## A    Flowchart of the proofs

Although the proofs of our theorems are intricate, many lemmas have clear intuition behind the statement. Therefore, we add "*" to these lemmas, so that time constrained readers could feel confident to skip the proofs. We also plot a flowchart of the proofs in Figure 9 to help the readers spend time wisely.

Since the proofs are long and complicated, we choose to present them in a top-down way. That is, we present the main theorems (Theorem 3.1, Theorem 3.2, and Theorem 3.3) in the main paper, and then present the necessary lemmas in order to prove those main theorems in Section B, Section C and Section D. Finally, we present the proofs for those lemma in Section G, Section H and Section I, respectively.

## B    Compute Approximation Matrix

The exact form of $-\nabla \mathsf{L}(\mathbf{W})_j$ in Lemma 2.1 contains variables like $\theta_{i^*,j}, \theta_{i,j}, \sin\theta_{i^*,j}, \sin\theta_{i,j}$, which are hard to deal with. In this section, we compute the approximation of these terms using Taylor series, and show that the approximation loss is minor. While the proofs are technically involved, the claims themselves are not surprising. Hence, we encourage the readers to skip the proofs (Appendix G) for the first reading.

Define the $j$-th column of the approximation matrix $\mathbf{P}$ as follows. See Definition 2.2 and Definition 2.3 for $g_j, \mathbf{A}_j$.

$$\mathbf{P}_j \triangleq \mathbf{P}_{1,j} + \mathbf{P}_{2,j} + \mathbf{P}_{3,j}, \quad \text{where}$$

$$\mathbf{P}_{1,j} \triangleq \sum_{i=1}^{d} \frac{\pi}{2}(w_i^* - w_i),$$

$$\mathbf{P}_{2,j} \triangleq g_j \overline{e_j + w_j} + \left(\mathbf{I} - \frac{1}{2}\overline{e_j + w_j} \cdot \overline{e_j + w_j}^\top\right)\mathbf{A}_j \overline{e_j + w_j},$$

$$\mathbf{P}_{3,j} \triangleq \left(\frac{\pi}{2} - \theta_{j^*,j}\right)(e_j + w_j^*) - \frac{\pi}{2}(e_j + w_j) + \|e_j + w_j^*\| \sin\theta_{j^*,j} \overline{e_j + w_j}.$$

Treat $\mathbf{P}_{1,j}, \mathbf{P}_{2,j}, \mathbf{P}_{3,j}$ as $j$-th column of matrix $\mathbf{P}_1, \mathbf{P}_2, \mathbf{P}_3$ respectively, we have $\mathbf{P} = \mathbf{P}_1 + \mathbf{P}_2 + \mathbf{P}_3$. Although $\mathbf{P}$ depends on $\mathbf{W}$, we abuse the notation and simply write $\mathbf{P}$.

**Claim B.1.** $\mathbf{P}_j$ approximates $-\nabla\mathsf{L}(\mathbf{W})_j$ by setting $(\frac{\pi}{2} - \theta_{i,j}) \approx \langle \overline{e_i + w_i}, \overline{e_j + w_j}\rangle$, $(\frac{\pi}{2} - \theta_{i^*,j}) \approx \langle \overline{e_i + w_i^*}, \overline{e_j + w_j}\rangle$, $\sin\theta_{i,j} \approx 1 - \frac{1}{2}\langle \overline{e_i + w_i}, \overline{e_j + w_j}\rangle^2$ and $\sin\theta_{i^*,j} \approx 1 - \frac{1}{2}\langle \overline{e_i + w_i^*}, \overline{e_j + w_j}\rangle^2$.

Below we show that the approximation loss is negligible in terms of one point convexity and spectral norm.

**Lemma\* B.2.** If $\|\mathbf{W}\|_2, \|\mathbf{W}^*\|_2 \le \gamma \le \frac{1}{100}$, $|\langle \mathbf{P} + \nabla\mathsf{L}(\mathbf{W}), \mathbf{W}^* - \mathbf{W}\rangle| < 0.085\|\mathbf{W}^* - \mathbf{W}\|_F^2$.

**Lemma\* B.3.** If $\|\mathbf{W}\|_2, \|\mathbf{W}^*\|_2 \le \gamma \le \frac{1}{100}$, $\|\mathbf{P} + \nabla\mathsf{L}(\mathbf{W})\|_2 \le 3.5\gamma^2$.

## C    Phase I: The Decreasing Potential Function

As we saw in Theorem 3.3, if $\|\mathbf{W}\|_2, \|\mathbf{W}^*\|_2$ is bounded by a constant $\gamma = \frac{1}{100}$, and the potential function $g \leq 0.1$, $\mathsf{L}(\mathbf{W})$ is 0.03-one point convex, which will give us convergence guarantee according to Lemma 2.5. However, $g$ could be larger than 0.1 initially, and as we run SGD, $\|\mathbf{W}\|_2$ might be larger than $\frac{1}{100}$ as well.

In this section, we address both problems by analyzing the dynamics of SGD, thus prove Theorem 3.2. The proofs can be found in Appendix H. Before proceeding to the interesting stuff, we need a simpler form of $\nabla \mathsf{L}(\mathbf{W})$ to work with, see below.

**Lemma C.1.** *If* $\|\mathbf{W}\|_2, \|\mathbf{W}^*\|_2 \leq \gamma \leq \frac{1}{100}$, *the negative gradient of* $\mathsf{L}(\mathbf{W})$ *is approximately*

$$\mathbf{Q}(\mathbf{W}) \triangleq \frac{\pi}{2}(\mathbf{W}^* - \mathbf{W})\left(\mathbf{I} + uu^\top\right) + (\mathbf{W}^* - \mathbf{W})^\top - 2\mathrm{Diag}(\mathbf{W}^* - \mathbf{W}) + g\overline{\mathbf{I} + \mathbf{W}}$$

*where $u$ is the all 1 vector. The approximation error is* $\|\mathbf{Q}(\mathbf{W}) - [-\nabla \mathsf{L}(\mathbf{W})]\|_2 \leq 61\gamma^2$.

We immediately get the bound of the gradient norm.

**Lemma\* C.2.** *If* $\|\mathbf{W}\|_2, \|\mathbf{W}^*\|_2 \leq \gamma \leq \frac{1}{100}$, $\|\nabla \mathsf{L}(\mathbf{W})\|_2 \leq 6d\gamma$.

Now we are ready to analyze the dynamics. We use subscript $t$ under each variable to denote its value at the step $t$. For simplicity, let $\mathbf{Q}_t \triangleq \mathbf{Q}(\mathbf{W}_t)$. Define $s_t \triangleq (\mathbf{W}^* - \mathbf{W}_t)u$. We first compute the updating rule for $g_t$.

**Lemma C.3.** *If* $\|\mathbf{W}_t\|_2, \|\mathbf{W}^*\|_2 \leq \gamma \leq \frac{1}{100}$, $d \geq 100$, $\eta \leq \frac{\gamma^2}{G_2^2}$, *then* $|g_{t+1}| \leq (1 - 0.95\eta d)|g_t| + 86\eta d\gamma^2 + 1.03\eta\sqrt{d}\varepsilon + 4.8\eta\|s_t\|_2\gamma\sqrt{d}$.

The bound contains $\|s_t\|_2$ which could be large, so we also need to compute its updating rule:

**Lemma C.4.** *If* $\|\mathbf{W}_t\|_2, \|\mathbf{W}^*\|_2 \leq \gamma \leq \frac{1}{100}$, *then* $\|s_{t+1}\|_2 \leq \left(1 - \eta\frac{(d+1)\pi}{2}\right)\|s_t\|_2 + \eta(6.61\gamma + 1.03|g_t| + \varepsilon)\sqrt{d}$.

Combining the two lemmas, we are ready to show that $g_t$ will shrink, conditioned on that $\|\mathbf{W}_t\|_2$ is bounded by $\gamma$.

**Lemma C.5.** *If for every step* $t > 0$, $\|\mathbf{W}_t\|_2, \|\mathbf{W}^*\|_2 \leq \gamma \leq \frac{1}{100}$, $d \geq 100$, $\eta \leq \frac{\gamma^2}{G_2^2}$, $\varepsilon \leq \gamma^2$, *then* $|g_t|$ *will keep decreasing by a factor of* $1 - 0.5\eta d$ *for every step, until* $|g_{t_1}| \leq 197\gamma^2$ *for* $t_1 \leq \frac{1}{16\eta}$.

Fortunately, we also know that $\|\mathbf{W}_t\|_2$ is always bounded by $\gamma$ during the process described in Lemma C.5.

**Lemma C.6.** *There exists a constant* $\gamma > \gamma_0 > 0$ *such that if* $\|\mathbf{W}_0\|_2, \|\mathbf{W}^*\|_2 \leq \gamma_0$, $d \geq 100$, $\eta \leq \frac{\gamma^2}{G_2^2}$, $\varepsilon \leq \gamma^2$, *then in the process of Phase I (Lemma C.5), we always have* $\|\mathbf{W}_T\|_2 \leq \gamma \leq \frac{1}{100}$ *for any* $T > 0$.

Now, we are at the state where $|g_t|$ is small, and $\|\mathbf{W}_T\|_2 \leq \gamma$, which means we are in Phase II. The next lemma ensures that we will stay in Phase II forever.

**Lemma C.7.** *There exists a constant* $\gamma_0 > \gamma > 0$ *such that if* $\|\mathbf{W}_0\|_2, \|\mathbf{W}^*\|_2 \leq \gamma_0$, $d \geq 100$, $\eta \leq \frac{\gamma^2}{G_2^2}$, $\varepsilon \leq \gamma^2$, *then after* $|g_{t_1}| \leq 197\gamma^2$, *Phase I ends and Phase II starts. That is, for every* $T > t_1$, $\|\mathbf{W}_T\|_2 \leq \gamma$ *and* $|g_T| \leq 0.1$.

*Proof for Theorem 3.2.* We immediately get Theorem 3.2 by combining the above three lemmas. They show that $g_t$ will decrease to a small value in Phase I (Lemma C.5), $\|\mathbf{W}_t\|_2$ will keep small during this process (Lemma C.6), and they all keep small afterwards (Lemma C.7). $\quad\square$

## D    Phase II: One Point Convexity

In this section, we prove Theorem 3.3. See detailed proofs in Appendix I. Using Lemma B.2, it suffices to bound

$$\langle \mathbf{P}, \mathbf{W}^* - \mathbf{W} \rangle = \sum_{j=1}^{d} \langle \mathbf{P}_{1,j} + \mathbf{P}_{2,j} + \mathbf{P}_{3,j}, w_j^* - w_j \rangle$$

Here the first term is easy to calculate.

$$\sum_{j=1}^{d}\langle \mathbf{P}_{1,j}, w_j^* - w_j\rangle = \frac{\pi}{2}\left\|\sum_{i=1}^{d}(w_i^* - w_i)\right\|_2^2 \geq 0 \tag{3}$$

For notational simplicity, denote

$$x_j \triangleq \left(\overline{e_j + w_j} \cdot \overline{e_j + w_j}^\top\right)(w_j^* - w_j),$$
$$\mathbf{X} \triangleq (x_1, \cdots, x_d) \tag{4}$$
$$z_j \triangleq \left(\mathbf{I} - \frac{1}{2}\overline{e_j + w_j} \cdot \overline{e_j + w_j}^\top\right)(w_j^* - w_j) \tag{5}$$

By Definition of $\mathbf{P}_{2,j}$ and (5), we have

$$\sum_{j=1}^{d}\langle \mathbf{P}_{2,j}, w_j^* - w_j\rangle = \sum_{j=1}^{d}\langle g_j\overline{e_j + w_j}, w_j^* - w_j\rangle + \sum_{j=1}^{d}z_j^\top \mathbf{A}_j\overline{e_j + w_j} \tag{6}$$

We bound the above two terms separately below.

**Lemma D.1.** *If* $\|\mathbf{W}\|_2, \|\mathbf{W}^*\|_2 \leq \gamma \leq \frac{1}{100}$, *then*

$$\sum_{j=1}^{d}z_j^\top \mathbf{A}_j\overline{e_j + w_j} \geq -(1.3 + 8\gamma)\|\mathbf{W}^* - \mathbf{W}\|_F^2 + \|\mathbf{W}^* - \mathbf{W}\|_F\|\mathbf{X}\|_F.$$

**Lemma D.2.** *If* $\|\mathbf{W}\|_2, \|\mathbf{W}^*\|_2 \leq \gamma \leq \frac{1}{100}$, *then*

$$\sum_{j=1}^{d}\langle g_j\overline{e_j + w_j}, w_j^* - w_j\rangle \geq -\|\mathbf{W}^* - \mathbf{W}\|_F\|\mathbf{X}\|_F - \frac{(1+\gamma)g\|\mathbf{W}^* - \mathbf{W}\|_F^2}{2(1-2\gamma)}$$

It remains to bound $\sum_{j=1}^{d}\langle \mathbf{P}_{3,j}, w_j^* - w_j\rangle$. We have the following lemma.

**Lemma D.3.** *If* $\|\mathbf{W}\|_2, \|\mathbf{W}^*\|_2 \leq \gamma \leq \frac{1}{100}$, $\sum_{j=1}^{d}\langle \mathbf{P}_{3,j}, w_j^* - w_j\rangle \geq \left(\frac{\pi}{2} - 0.021\right)\|\mathbf{W}^* - \mathbf{W}\|_F^2$.

*Proof of Theorem 3.3.* By (3), (6), Lemma D.1, Lemma D.2 and Lemma D.3, we know

$$\langle \mathbf{P}, \mathbf{W}^* - \mathbf{W}\rangle \geq \left(\frac{\pi}{2} - 1.321 - 8\gamma - \frac{(1+\gamma)g}{2(1-2\gamma)}\right)\|\mathbf{W}^* - \mathbf{W}\|_F^2 > \left(0.169 - \frac{(1+\gamma)g}{2(1-2\gamma)}\right)\|\mathbf{W}^* - \mathbf{W}\|_F^2$$

Using Lemma B.2, we get

$$\langle -\nabla\mathsf{L}(\mathbf{W}), \mathbf{W}^* - \mathbf{W}\rangle > \left(0.084 - \frac{(1+\gamma)g}{2(1-2\gamma)}\right)\|\mathbf{W}^* - \mathbf{W}\|_F^2 > 0.03\|\mathbf{W}^* - \mathbf{W}\|_F^2$$

The last inequality holds when $g \leq 0.1$. $\qquad\square$

## E  A Geometric Lemma

In our proof, we need very tight bounds for a few terms. In order to get such bounds, we present a nice and intuitive geometric lemma as follows.

**Lemma E.1.** *If* $\|\mathbf{W}\|_2, \|\mathbf{W}^*\|_2 \leq \gamma$, *then* $\forall i \in [d]$,

1. $\|\overline{e_i + w_i^*} - \overline{e_i + w_i}\|_2 \leq \frac{\|(\mathbf{I} - \overline{e_i + w_i} \cdot \overline{e_i + w_i}^\top)(w_i^* - w_i)\|_2}{\sqrt{1-2\gamma}} \leq \frac{\|w_i^* - w_i\|_2}{\sqrt{1-2\gamma}}$

2. $-\frac{\|w_i^* - w_i\|_2^2}{2(1-2\gamma)} \leq \langle \overline{e_i + w_i^*} - \overline{e_i + w_i}, \overline{e_i + w_i}\rangle \leq 0$

3. *if* $\gamma \leq \frac{1}{100}$, $0 \leq \theta_{i,i^*} \leq 1.001\|w_i^* - w_i\|_2$.

Figure 10: For Lemma E.1

*Proof.* See Figure 10. Denote $e_i + w_i^*$ as $\overrightarrow{OC}$, $e_i + w_i$ as $\overrightarrow{OD}$, $\overline{e_i + w_i^*}$ as $\overrightarrow{OA}$, $\overline{e_i + w_i}$ as $\overrightarrow{OB}$. Thus, $\|w_i^* - w_i\|_2 = \|\overrightarrow{DC}\|_2$.

**1.** Since $\overrightarrow{OD} \perp \overrightarrow{CF}$, we know $\|\overrightarrow{CD}\|_2 \geq \|\overrightarrow{CF}\|_2$. Since $\triangle CFO \sim \triangle AEO$, we know

$$\frac{\|\overrightarrow{CD}\|_2}{\|\overrightarrow{AE}\|_2} \geq \frac{\|\overrightarrow{CF}\|_2}{\|\overrightarrow{AE}\|_2} = \frac{\|\overrightarrow{OC}\|_2}{\|\overrightarrow{OA}\|_2} = \|e_i + w_i^*\|_2 \geq 1 - \gamma \tag{7}$$

The last inequality holds as $\|\mathbf{W}^*\|_2 \leq \gamma$.

Notice that $\|\overrightarrow{OA}\|_2 = \|\overrightarrow{OB}\|_2 = 1$, we know $\triangle ABO$ is a isosceles triangle. Thus, $\|\overrightarrow{AG}\|_2 = \|\overrightarrow{GB}\|_2$. Notice that $\triangle ABE \sim \triangle BGO$, we have

$$\frac{\|\overrightarrow{AE}\|_2}{\|\overrightarrow{AB}\|_2} = \frac{\|\overrightarrow{OG}\|_2}{\|\overrightarrow{OB}\|_2} = \frac{\sqrt{1 - \|\overrightarrow{GB}\|_2^2}}{1} \tag{8}$$

WLOG, assume $\|\overrightarrow{OC}\|_2 \geq \|\overrightarrow{OD}\|_2$, as shown in the figure. We draw $\overrightarrow{HB} \parallel \overrightarrow{CD}$, and we know $\|\overrightarrow{OH}\|_2 \geq \|\overrightarrow{OB}\|_2 = \|\overrightarrow{OA}\|_2$. Since $\triangle CDO \sim \triangle HBO$, we have

$$\frac{\|\overrightarrow{CD}\|_2}{\|\overrightarrow{HB}\|_2} = \frac{\|\overrightarrow{OD}\|_2}{\|\overrightarrow{OB}\|_2} = \|\overrightarrow{OD}\|_2 \geq 1 - \gamma$$

So $\|\overrightarrow{CD}\|_2 \geq (1 - \gamma)\|\overrightarrow{HB}\|_2$. On the other hand, $\angle BAO < \frac{\pi}{2}$, and $A$ is between $H$ and $O$, so $\angle BAH > \frac{\pi}{2}$, which means $\|\overrightarrow{HB}\|_2 \geq \|\overrightarrow{AB}\|_2 = 2\|\overrightarrow{GB}\|_2$. Thus, $\|\overrightarrow{GB}\|_2 \leq \frac{\|\overrightarrow{HB}\|_2}{2} \leq \frac{\|\overrightarrow{CD}\|_2}{2(1-\gamma)}$.

Substitute it into (8), we get

$$\frac{\|\overrightarrow{AE}\|_2}{\|\overrightarrow{AB}\|_2} \geq \sqrt{1 - \frac{\|\overrightarrow{CD}\|_2^2}{4(1-\gamma)^2}} \geq \sqrt{1 - \left(\frac{\gamma}{1-\gamma}\right)^2}$$

The last inequality holds since $\|\overrightarrow{CD}\|_2 = \|w_i^* - w_i\|_2 \leq 2\gamma$.

Substitute this inequality into (7), we get

$$\|\overline{e_i + w_i^*} - \overline{e_i + w_i}\|_2 = \|\overrightarrow{AB}\|_2$$

$$\leq \frac{\|\overrightarrow{AE}\|_2}{\sqrt{1 - \left(\frac{\gamma}{1-\gamma}\right)^2}} \leq \frac{\|\overrightarrow{CF}\|_2}{(1-\gamma)\sqrt{1 - \left(\frac{\gamma}{1-\gamma}\right)^2}} \tag{9}$$

$$\leq \frac{\|\overrightarrow{CD}\|_2}{(1-\gamma)\sqrt{1 - \left(\frac{\gamma}{1-\gamma}\right)^2}} = \frac{\|w_i^* - w_i\|_2}{\sqrt{1 - 2\gamma}} \tag{10}$$

Notice that $\overline{e_i + w_i}^\top (w_i^* - w_i) = -\|\overrightarrow{DF}\|_2$, so $\overline{e_i + w_i} \cdot \overline{e_i + w_i}^\top (w_i^* - w_i) = \overrightarrow{DF}$. That means,

$$\|(\mathbf{I} - \overline{e_i + w_i} \cdot \overline{e_i + w_i}^\top)(w_i^* - w_i)\|_2 = \|\overrightarrow{DC} - \overrightarrow{DF}\|_2 = \|\overrightarrow{CF}\|_2$$

The lemma follows by (9) and (10).

**2.** By Figure 10, we know $|\langle \overline{e_i + w_i^*} - \overline{e_i + w_i}, \overline{e_i + w_i}\rangle| = \|\overrightarrow{BE}\|_2$. Since $\triangle ABE \sim \triangle GBO$, we have

$$\frac{\|\overrightarrow{BE}\|_2}{\|\overrightarrow{AB}\|_2} = \frac{\|\overrightarrow{GB}\|_2}{\|\overrightarrow{BO}\|_2} = \frac{\|\overrightarrow{AB}\|_2}{2}$$

Therefore, using (10) we get

$$|\langle \overline{e_i + w_i^*} - \overline{e_i + w_i}, \overline{e_i + w_i}\rangle| = \frac{\|\overrightarrow{AB}\|_2^2}{2} \leq \frac{\|w_i^* - w_i\|_2^2}{2(1 - 2\gamma)}$$

Moreover, $\langle \overline{e_i + w_i^*} - \overline{e_i + w_i}, \overline{e_i + w_i}\rangle = \langle \overline{e_i + w_i^*}, \overline{e_i + w_i}\rangle - 1 \leq 0$.

**3.** We know that

$$\theta_{i,i^*} = 2\arcsin \|\overrightarrow{AG}\|_2 = 2\arcsin \frac{\|\overline{e_i + w_i^*} - \overline{e_i + w_i}\|_2}{2}$$

$$\leq \|\overline{e_i + w_i^*} - \overline{e_i + w_i}\|_2 + \frac{\|\overline{e_i + w_i^*} - \overline{e_i + w_i}\|_2^3}{8}$$

The last inequality holds by Taylor's Series for $\arcsin$, and the fact $\|\overline{e_i + w_i^*} - \overline{e_i + w_i}\|_2 = \|\overrightarrow{AB}\|_2 \leq \|w_i^* - w_i\|_2 \leq 2\gamma \leq \frac{1}{50}$. Thus, we have $\theta_{i,i^*} \leq 1.001\|w_i^* - w_i\|_2$. $\qquad\square$

# F    More Handy Lemmas

**Lemma* F.1.** *If* $\|\mathbf{W}\|_2, \|\mathbf{W}^*\|_2 \leq \gamma$, *then*

- $\frac{(1-\gamma)^2}{(1+\gamma)^2}\mathbf{I} \preceq \overline{\mathbf{I} + \mathbf{W}}^\top \overline{\mathbf{I} + \mathbf{W}} \preceq \frac{(1+\gamma)^2}{(1-\gamma)^2}\mathbf{I}, \quad \frac{(1-\gamma)^2}{(1+\gamma)^2}\mathbf{I} \preceq \overline{\mathbf{I} + \mathbf{W}^*}^\top \overline{\mathbf{I} + \mathbf{W}^*} \preceq \frac{(1+\gamma)^2}{(1-\gamma)^2}\mathbf{I},$

- $(1 - \gamma)^2\mathbf{I} \preceq (\mathbf{I} + \mathbf{W})^\top(\mathbf{I} + \mathbf{W}) \preceq (1 + \gamma)^2\mathbf{I}, \quad (1 - \gamma)^2\mathbf{I} \preceq (\mathbf{I} + \mathbf{W}^*)^\top(\mathbf{I} + \mathbf{W}^*) \preceq (1 + \gamma)^2\mathbf{I}.$

*Therefore, the singular value of* $\overline{\mathbf{I} + \mathbf{W}}$ *is at most* $\frac{1+\gamma}{1-\gamma}$ *and at least* $\frac{1-\gamma}{1+\gamma}$. *The singular value of* $\mathbf{I} + \mathbf{W}$ *is at most* $1 + \gamma$ *and at least* $1 - \gamma$. *The same claims hold for* $\overline{\mathbf{I} + \mathbf{W}^*}$, $\mathbf{I} + \mathbf{W}^*$ *respectively.*

*Proof.* Since $\|\mathbf{W}\|_2 \leq \gamma$, we have $1 - \gamma \leq \|\mathbf{I} + \mathbf{W}\|_2 \leq 1 + \gamma$, and $1 - \gamma \leq \|e_i + w_i\|_2 \leq 1 + \gamma$. Therefore, $\overline{\mathbf{I} + \mathbf{W}} = \mathbf{\Sigma}(\mathbf{I} + \mathbf{W})$ where $\mathbf{\Sigma}$ is a diagonal matrix whose entries are within $[\frac{1}{1+\gamma}, \frac{1}{1-\gamma}]$. Putting into $\overline{\mathbf{I} + \mathbf{W}}^\top \overline{\mathbf{I} + \mathbf{W}}$, we have

$$\overline{\mathbf{I} + \mathbf{W}}^\top \overline{\mathbf{I} + \mathbf{W}} = (\mathbf{I} + \mathbf{W})^\top \mathbf{\Sigma}^2(\mathbf{I} + \mathbf{W}) \preceq \frac{1}{(1 - \gamma)^2}(\mathbf{I} + \mathbf{W})^\top(\mathbf{I} + \mathbf{W}) \preceq \frac{(1 + \gamma)^2}{(1 - \gamma)^2}\mathbf{I}$$

Similarly we can show $\overline{\mathbf{I} + \mathbf{W}}^\top \overline{\mathbf{I} + \mathbf{W}} \succeq \frac{(1-\gamma)^2}{(1+\gamma)^2}\mathbf{I}$. Thus we know the singular value of $\overline{\mathbf{I} + \mathbf{W}}$ is at most $\frac{1+\gamma}{1-\gamma}$ and at least $\frac{1-\gamma}{1+\gamma}$. The same proof works for $\mathbf{I} + \mathbf{W}, \overline{\mathbf{I} + \mathbf{W}^*}$ and $\mathbf{I} + \mathbf{W}^*$. $\qquad\square$

**Lemma* F.2.** *If* $\|\mathbf{W}\|_2, \|\mathbf{W}^*\|_2 \leq \gamma \leq \frac{1}{100}$, *we have*

$$|\langle \overline{e_i + w_i^*}, \overline{e_j + w_j}\rangle| \leq 2.1\gamma, \quad |\langle \overline{e_i + w_i}, \overline{e_j + w_j}\rangle| \leq 2.1\gamma$$

*Proof.* We know

$$|\langle \overline{e_i + w_i^*}, \overline{e_j + w_j}\rangle| = \frac{|\langle e_i + w_i^*, e_j + w_j\rangle|}{\|e_i + w_i^*\|_2\|e_j + w_j\|_2} \leq \frac{|\langle e_i + w_i^*, e_j + w_j\rangle|}{(1 - \gamma)^2} = \frac{|w_{i,j}^*| + |w_{i,j}| + |\langle w_i, w_j\rangle|}{(1 - \gamma)^2} \leq \frac{(2 + \gamma)\gamma}{(1 - \gamma)^2} \leq 2.1\gamma$$

where the last inequality holds since $\gamma \leq \frac{1}{100}$. The same analysis works for $\langle \overline{e_i + w_i}, \overline{e_j + w_j}\rangle$. $\quad\square$

**Lemma* F.3** (Triangle inequality between $e_i + w_i, e_i + w_i^*, w_i^* - w_i$). $|\|e_i + w_i\|_2 - \|e_i + w_i^*\|_2| \leq \|w_i^* - w_i\|_2$.

**Lemma\* F.4.** *If* $\|\mathbf{W}\|_2, \|\mathbf{W}^*\|_2 \le \gamma$, $|g| \le 2d\gamma$.

*Proof.* By definition and Lemma F.3, we know $|g| = \sum_{i=1}^d (\|e_i + w_i^*\|_2 - \|e_i + w_i\|_2) \le \sum_{i=1}^d \|w_i^* - w_i\|_2 \le 2d\gamma$. $\qquad\square$

**Lemma\* F.5.** *If* $\|\mathbf{W}\|_2, \|\mathbf{W}^*\|_2 \le \gamma$, $|\langle \overline{e_i + w_i^*} - \overline{e_i + w_i}, \overline{e_j + w_j} \rangle| \le \frac{\|w_i^* - w_i\|_2}{\sqrt{1 - 2\gamma}}$.

*Proof.* By Cauchy Schwartz and Lemma E.1 term 1. $\qquad\square$

**Lemma\* F.6.** $|x^k - y^k| \le \frac{k}{2}|x - y|(|x|^{k-1} + |y|^{k-1})$.

*Proof.* $|x^k - y^k| = \left|(x - y)\sum_{t=1}^{k-1}\frac{x^t y^{k-t-1} + y^t x^{k-t-1}}{2}\right| \le \frac{k}{2}|x - y|(|x|^{k-1} + |y|^{k-1})$, where the last inequality holds since $|x^t y^{k-t-1} + y^t x^{k-t-1}| \le |x|^t |y|^{k-t-1} + |y|^t |x|^{k-t-1} \le |x|^{k-1} + |y|^{k-1}$, by rearrangement inequality. $\qquad\square$

**Lemma\* F.7.** *If* $\|\mathbf{W}\|_2, \|\mathbf{W}^*\|_2 \le \gamma \le \frac{1}{100}$, *for* $k \ge 3$, *we have*

$$\|\langle \overline{e_i + w_i^*}, \overline{e_j + w_j} \rangle^k (e_i + w_i^*) - \langle \overline{e_i + w_i}, \overline{e_j + w_j} \rangle^k (e_i + w_i)\|_2$$
$$\le 6(2.2\gamma)^{k-3} \left( \langle \overline{e_i + w_i^*}, \overline{e_j + w_j} \rangle^2 + \langle \overline{e_i + w_i}, \overline{e_j + w_j} \rangle^2 \right) \|w_i^* - w_i\|_2$$

*Proof.*

$$\|\langle \overline{e_i + w_i^*}, \overline{e_j + w_j} \rangle^k (e_i + w_i^*) - \langle \overline{e_i + w_i}, \overline{e_j + w_j} \rangle^k (e_i + w_i)\|_2$$
$$\le \|w_i^* - w_i\|_2 |\langle \overline{e_i + w_i^*}, \overline{e_j + w_j} \rangle^k| + \|(\langle \overline{e_i + w_i^*}, \overline{e_j + w_j} \rangle^k - \langle \overline{e_i + w_i}, \overline{e_j + w_j} \rangle^k)(e_i + w_i)\|_2$$
$$\le \|w_i^* - w_i\|_2 |\langle \overline{e_i + w_i^*}, \overline{e_j + w_j} \rangle^k| + (1 + \gamma)|\langle \overline{e_i + w_i^*}, \overline{e_j + w_j} \rangle^k - \langle \overline{e_i + w_i}, \overline{e_j + w_j} \rangle^k|$$
$$\overset{①}{\le} \|w_i^* - w_i\|_2 (2.1\gamma)^{k-2} \langle \overline{e_i + w_i^*}, \overline{e_j + w_j} \rangle^2$$
$$\quad + \frac{(1 + \gamma)k}{2} |\langle \overline{e_i + w_i^*} - \overline{e_i + w_i}, \overline{e_j + w_j} \rangle|(|\langle \overline{e_i + w_i^*}, \overline{e_j + w_j} \rangle|^{k-1} + |\langle \overline{e_i + w_i}, \overline{e_j + w_j} \rangle|^{k-1})$$
$$\le \langle \overline{e_i + w_i^*}, \overline{e_j + w_j} \rangle^2 \left( \|w_i^* - w_i\|_2 (2.1\gamma)^{k-2} + \frac{(1 + \gamma)k(2.1\gamma)^{k-3}}{2}|\langle \overline{e_i + w_i^*} - \overline{e_i + w_i}, \overline{e_j + w_j} \rangle| \right)$$
$$\quad + \langle \overline{e_i + w_i}, \overline{e_j + w_j} \rangle^2 \left( \frac{(1 + \gamma)k(2.1\gamma)^{k-3}}{2}|\langle \overline{e_i + w_i^*} - \overline{e_i + w_i}, \overline{e_j + w_j} \rangle| \right)$$
$$\overset{②}{\le} \|w_i^* - w_i\|_2 \left[ \left((2.1\gamma)^{k-2} + 0.52k(2.1\gamma)^{k-3}\right) \langle \overline{e_i + w_i^*}, \overline{e_j + w_j} \rangle^2 + 0.52k(2.1\gamma)^{k-3} \langle \overline{e_i + w_i}, \overline{e_j + w_j} \rangle^2 \right]$$
$$\overset{③}{\le} \|w_i^* - w_i\|_2 \left[ 0.55k(2.1\gamma)^{k-3} \langle \overline{e_i + w_i^*}, \overline{e_j + w_j} \rangle^2 + 0.52k(2.1\gamma)^{k-3} \langle \overline{e_i + w_i}, \overline{e_j + w_j} \rangle^2 \right]$$
$$\overset{④}{\le} 6(2.2\gamma)^{k-3} \left( \langle \overline{e_i + w_i^*}, \overline{e_j + w_j} \rangle^2 + \langle \overline{e_i + w_i}, \overline{e_j + w_j} \rangle^2 \right) \|w_i^* - w_i\|_2$$

where ① uses Lemma F.2 and Lemma F.6, ② uses Lemma F.5, ③ holds as $\gamma \le \frac{1}{100}$, and ④ holds since $0.55k(2.1)^{k-3} \le 6(2.2)^{k-3}$ for $k \ge 3$. $\qquad\square$

**Lemma\* F.8.** *If* $\|\mathbf{W}\|_2, \|\mathbf{W}^*\|_2 \le \gamma \le \frac{1}{100}$, *for* $k \ge 2$,

$$\left| \|e_i + w_i\|_2 \langle \overline{e_i + w_i}, \overline{e_j + w_j} \rangle^{2k} - \|e_i + w_i^*\|_2 \langle \overline{e_i + w_i^*}, \overline{e_j + w_j} \rangle^{2k} \right|$$
$$\le 8(2.2\gamma)^{2k-3} \left( \langle \overline{e_i + w_i}, \overline{e_j + w_j} \rangle^2 + \langle \overline{e_i + w_i^*}, \overline{e_j + w_j} \rangle^2 \right) \|w_i^* - w_i\|_2$$

*Proof.*

$$\left| \|e_i + w_i\|_2 \langle \overline{e_i + w_i}, \overline{e_j + w_j} \rangle^{2k} - \|e_i + w_i^*\|_2 \langle \overline{e_i + w_i^*}, \overline{e_j + w_j} \rangle^{2k} \right|$$

$$\leq \|e_i + w_i\|_2 \left| \langle \overline{e_i + w_i}, \overline{e_j + w_j} \rangle^{2k} - \langle \overline{e_i + w_i^*}, \overline{e_j + w_j} \rangle^{2k} \right| + \left| \|e_i + w_i\|_2 - \|e_i + w_i^*\|_2 \right| \langle \overline{e_i + w_i^*}, \overline{e_j + w_j} \rangle^{2k}$$

$$\overset{①}{\leq} \|e_i + w_i\|_2 \left| \langle \overline{e_i + w_i}, \overline{e_j + w_j} \rangle^{2k} - \langle \overline{e_i + w_i^*}, \overline{e_j + w_j} \rangle^{2k} \right| + \|w_i^* - w_i\|_2 (2.1\gamma)^{2k-2} \langle \overline{e_i + w_i^*}, \overline{e_j + w_j} \rangle^2$$

$$\overset{②}{\leq} (1+\gamma)k |\langle \overline{e_i + w_i} - \overline{e_i + w_i^*}, \overline{e_j + w_j} \rangle| \left( |\langle \overline{e_i + w_i}, \overline{e_j + w_j} \rangle|^{2k-1} + |\langle \overline{e_i + w_i^*}, \overline{e_j + w_j} \rangle|^{2k-1} \right)$$

$$\quad + \|w_i^* - w_i\|_2 (2.1\gamma)^{2k-2} \langle \overline{e_i + w_i^*}, \overline{e_j + w_j} \rangle^2$$

$$\overset{③}{\leq} \left[ \frac{(1+\gamma)k(2.1\gamma)^{2k-3}}{\sqrt{1-2\gamma}} \langle \overline{e_i + w_i}, \overline{e_j + w_j} \rangle^2 + \left( \frac{(1+\gamma)k(2.1\gamma)^{2k-3}}{\sqrt{1-2\gamma}} + (2.1\gamma)^{2k-2} \right) \langle \overline{e_i + w_i^*}, \overline{e_j + w_j} \rangle^2 \right] \|w_i^* - w_i\|_2$$

$$\overset{④}{\leq} 1.05k(2.1\gamma)^{2k-3} \left( \langle \overline{e_i + w_i}, \overline{e_j + w_j} \rangle^2 + \langle \overline{e_i + w_i^*}, \overline{e_j + w_j} \rangle^2 \right) \|w_i^* - w_i\|_2$$

$$\overset{⑤}{\leq} 8(2.2\gamma)^{2k-3} \left( \langle \overline{e_i + w_i}, \overline{e_j + w_j} \rangle^2 + \langle \overline{e_i + w_i^*}, \overline{e_j + w_j} \rangle^2 \right) \|w_i^* - w_i\|_2$$

where ① uses Lemma F.2 and Lemma F.3, ② uses Lemma F.6, ③ uses Lemma F.5, ④ holds as $\gamma \leq \frac{1}{100}$, and ⑥ holds as $1.05k(2.1)^{2k-3} \leq 8(2.2)^{2k-3}$ for $k \geq 2$. □

**Lemma\* F.9.** *If* $\|\mathbf{W}\|_2, \|\mathbf{W}^*\|_2 \leq \gamma$, *for fixed* $j \in [d]$,

$$\sum_{i \neq j} \langle \overline{e_i + w_i}, \overline{e_j + w_j} \rangle^2 \leq \frac{4\gamma}{(1-\gamma)^2}, \qquad \sum_{i \neq j} \langle \overline{e_i + w_i^*}, \overline{e_j + w_j} \rangle^2 \leq \frac{4\gamma(1+\gamma)}{1-2\gamma}.$$

*Similarly, for fixed* $i \in [d]$,

$$\sum_{j \neq i} \langle \overline{e_i + w_i}, \overline{e_j + w_j} \rangle^2 \leq \frac{4\gamma}{(1-\gamma)^2}, \qquad \sum_{j \neq i} \langle \overline{e_i + w_i^*}, \overline{e_j + w_j} \rangle^2 \leq \frac{4\gamma(1+\gamma)}{1-2\gamma}.$$

*Proof.* By matrix multiplication,

$$\sum_{i=1}^d \langle \overline{e_i + w_i^*}, \overline{e_j + w_j} \rangle^2 = \sum_{i=1}^d \overline{e_j + w_j}^\top \overline{e_i + w_i^*} \cdot \overline{e_i + w_i^*}^\top \overline{e_j + w_j} = \overline{e_j + w_j}^\top \overline{\mathbf{I} + \mathbf{W}^*} \cdot \overline{\mathbf{I} + \mathbf{W}^*}^\top \overline{e_j + w_j}$$

By Lemma F.1, we know $\overline{\mathbf{I} + \mathbf{W}^*} \cdot \overline{\mathbf{I} + \mathbf{W}^*}^\top \preceq \frac{(1+\gamma)^2}{(1-\gamma)^2} \mathbf{I}$. That means, $\sum_{i=1}^d \langle \overline{e_i + w_i^*}, \overline{e_j + w_j} \rangle^2 \leq \frac{(1+\gamma)^2}{(1-\gamma)^2}$. On the other hand, by Lemma E.1 term 2, $\langle \overline{e_j + w_j^*}, \overline{e_j + w_j} \rangle^2 = (1 - \langle \overline{e_j + w_j^*} - \overline{e_j + w_j}, \overline{e_j + w_j} \rangle)^2 \geq 1 - \frac{\|w_i^* - w_i\|_2^2}{1-2\gamma}$.

Therefore, we know

$$\sum_{i \neq j} \langle \overline{e_i + w_i^*}, \overline{e_j + w_j} \rangle^2 \leq \frac{(1+\gamma)^2}{(1-\gamma)^2} - 1 + \frac{\|w_i^* - w_i\|_2^2}{1-2\gamma} = \frac{4\gamma}{(1-\gamma)^2} + \frac{\|w_i^* - w_i\|_2^2}{1-2\gamma} \leq \frac{4\gamma(1+\gamma)}{1-2\gamma}$$

Using the same analysis, we get $\sum_{i \neq j} \langle \overline{e_i + w_i}, \overline{e_j + w_j} \rangle^2 \leq \frac{(1+\gamma)^2}{(1-\gamma)^2} - 1 = \frac{4\gamma}{(1-\gamma)^2}$. The analysis for fixed $i$ is similar. □

**Lemma\* F.10.** *For any matrix* $\mathbf{A}$, *we have* $\|\mathrm{Diag}(\mathbf{A})\|_2 \leq \|\mathbf{A}\|_2$ *and* $\|\text{Off-Diag}(\mathbf{A})\|_2 \leq 2\|\mathbf{A}\|_2$.

*Proof.* By definition, we know $\|\mathrm{Diag}(\mathbf{A})\|_2 = \max_{i \in [d]} e_i^\top \mathbf{A} e_i \leq \max_{v \in \mathbb{R}^d} v^\top \mathbf{A} v = \|\mathbf{A}\|_2$, and $\|\text{Off-Diag}(\mathbf{A})\|_2 \leq \|\mathbf{A}\|_2 + \|\mathrm{Diag}(\mathbf{A})\|_2 \leq 2\|\mathbf{A}\|_2$. □

**Lemma\* F.11.** *If* $\|\mathbf{W}\|_2, \|\mathbf{W}^*\|_2 \leq \gamma$, $\|\mathbf{A}\|_2 \leq \frac{2\gamma(\gamma^2+3)}{1-\gamma^2}$.

*Proof.* By Lemma F.1, we have

$$\|\mathbf{A}\|_2 = \|(\mathbf{I} + \mathbf{W}^*)\overline{\mathbf{I} + \mathbf{W}^*}^\top - (\mathbf{I} + \mathbf{W})\overline{\mathbf{I} + \mathbf{W}}^\top\|_2 \leq \frac{(1+\gamma)^2}{1-\gamma} - \frac{(1-\gamma)^2}{1+\gamma} = \frac{2\gamma(\gamma^2+3)}{1-\gamma^2}. \quad □$$

**Lemma\* F.12.** *If* $\|\mathbf{W}\|_2, \|\mathbf{W}^*\|_2 \le \gamma \le \frac{1}{100}$, $|\overline{e_j + w_j}^\top \mathbf{A} \overline{e_j + w_j} - e_j^\top \mathbf{A} e_j| \le 5\gamma^2$.

*Proof.*

$$|\overline{e_j + w_j}^\top \mathbf{A} \overline{e_j + w_j} - e_j^\top \mathbf{A} e_j| \le |\overline{e_j + w_j}^\top \mathbf{A}(\overline{e_j + w_j} - e_j)| + |(\overline{e_j + w_j} - e_j)^\top \mathbf{A} e_j| \overset{①}{\le} \frac{4\gamma^2(\gamma^2 + 3)}{1 - \gamma^2} \overset{②}{<} 5\gamma^2$$

where ① uses Cauchy Schwartz, Lemma F.11 and $\|\overline{e_j + w_j} - e_j\|_2 \le \gamma$, and ② holds as $\gamma \le \frac{1}{100}$. $\qquad\square$

**Lemma\* F.13.** *For any* $i \in [n]$, $|[\|e_i + w_i^*\|_2 - \|e_i + w_i\|_2] - [w_{i,i}^* - w_{i,i}]| \le 6.07\gamma^2$.

*Proof.*

$$
\begin{aligned}
\|e_i + w_i\|_2 - \|e_i + w_i^*\|_2 &= \langle e_i + w_i, \overline{e_i + w_i}\rangle - \langle e_i + w_i^*, \overline{e_i + w_i^*}\rangle \\
&= \langle e_i + w_i, \overline{e_i + w_i} - \overline{e_i + w_i^*}\rangle + \langle w_i - w_i^*, \overline{e_i + w_i^*}\rangle \\
&= \langle w_i - w_i^*, e_i\rangle + \langle e_i + w_i, \overline{e_i + w_i} - \overline{e_i + w_i^*}\rangle + \langle w_i - w_i^*, \overline{e_i + w_i^*} - e_i\rangle \\
&= w_{i,i} - w_{i,i}^* + \langle e_i + w_i, \overline{e_i + w_i} - \overline{e_i + w_i^*}\rangle + \langle w_i - w_i^*, \overline{e_i + w_i^*} - e_i\rangle
\end{aligned}
$$

As a result,

$$|[\|e_i + w_i\|_2 - \|e_i + w_i^*\|_2] - [w_{i,i} - w_{i,i}^*]| \le |\langle e_i + w_i, \overline{e_i + w_i} - \overline{e_i + w_i^*}\rangle| + |\langle w_i - w_i^*, \overline{e_i + w_i^*} - e_i\rangle|$$

$$\overset{①}{\le} \frac{(1 + \gamma)2\gamma^2}{1 - 2\gamma} + 4\gamma^2 \le 6.07\gamma^2$$

where ① uses Lemma E.1 term 2 and $\|\overline{e_i + w_i^*} - e_i\|_2 \le 2\gamma$, and Cauchy Schwartz. So the claim follows. $\qquad\square$

**Corollary F.14.** $|g - \mathrm{Tr}(\mathbf{W}^* - \mathbf{W})| \le 6.07 d\gamma^2$.

**Lemma\* F.15.** $\overline{\mathbf{I} + \mathbf{W}}$ *is close to* $\mathbf{I}$ *on its diagonals, and close to* $\mathbf{W}$ *on its off-diagonals. More specifically, if* $\|\mathbf{W}\|_2, \|\mathbf{W}^*\|_2 \le \gamma \le \frac{1}{100}$,

$$\|\mathrm{Diag}(\overline{\mathbf{I} + \mathbf{W}}) - \mathbf{I}\|_2 \le \frac{\gamma^2}{2(1 - \gamma)^2}, \qquad \|\mathrm{Diag}(\overline{\mathbf{I} + \mathbf{W}^*}) - \mathbf{I}\|_2 \le \frac{\gamma^2}{2(1 - \gamma)^2}$$

$$\|\mathrm{Off\text{-}Diag}(\overline{\mathbf{I} + \mathbf{W}} - \mathbf{W})\|_2 \le \frac{4\gamma^2}{1 - \gamma}, \qquad \|\mathrm{Off\text{-}Diag}(\overline{\mathbf{I} + \mathbf{W}^*} - \mathbf{W}^*)\|_2 \le \frac{4\gamma^2}{1 - \gamma}$$

$$\|\overline{\mathbf{I} + \mathbf{W}} - \mathbf{I}\|_2 \le 2.05\gamma, \qquad \|\overline{\mathbf{I} + \mathbf{W}^*} - \mathbf{I}\|_2 \le 2.05\gamma$$

*Proof.* For the diagonal terms,

$$\|\mathrm{Diag}(\overline{\mathbf{I} + \mathbf{W}}) - \mathbf{I}\|_2 = \max_j |\overline{\mathbf{I} + \mathbf{W}}_{j,j} - 1| = \max_j \left| \frac{1 + w_{j,j} - \|e_j + w_j\|_2}{\|e_j + w_j\|_2} \right|$$

$$\le \max_j \left| \frac{(1 + w_{j,j})^2 - \|e_j + w_j\|_2^2}{\|e_j + w_j\|_2} \right| \left| \frac{1}{1 + w_{j,j} + \|e_j + w_j\|_2} \right| \le \max_j \frac{\sum_{i \ne j} w_{j,i}^2}{2(1 - \gamma)^2} \le \frac{\gamma^2}{2(1 - \gamma)^2}$$

For the off-diagonal terms, we know $\overline{\mathbf{I} + \mathbf{W}} = (\mathbf{I} + \mathbf{W})\boldsymbol{\Sigma}$ for some diagonal matrix $\boldsymbol{\Sigma}$, so

$$\|\mathrm{Off\text{-}Diag}(\overline{\mathbf{I} + \mathbf{W}} - \mathbf{W})\|_2 = \|\mathrm{Off\text{-}Diag}((\mathbf{I} + \mathbf{W})\boldsymbol{\Sigma} - \mathbf{W})\|_2 = \|\mathrm{Off\text{-}Diag}((\boldsymbol{\Sigma} - \mathbf{I})\mathbf{W})\|_2 \overset{①}{\le} 2\|(\boldsymbol{\Sigma} - \mathbf{I})\mathbf{W}\|_2 \le \frac{4\gamma^2}{1 - \gamma}$$

where ① uses Lemma F.10. For the difference between $\overline{\mathbf{I} + \mathbf{W}}$ and $\mathbf{I}$, we split $\overline{\mathbf{I} + \mathbf{W}}$ into diagonal and off-diagonal parts:

$$\|\overline{\mathbf{I} + \mathbf{W}} - \mathbf{I}\|_2 = \|\mathrm{Diag}(\overline{\mathbf{I} + \mathbf{W}}) + \mathrm{Off\text{-}Diag}(\overline{\mathbf{I} + \mathbf{W}}) - \mathbf{I}\|_2$$

$$= \|\mathrm{Off\text{-}Diag}(\mathbf{W})\|_2 + \frac{\gamma^2}{2(1 - \gamma)^2} + \frac{4\gamma^2}{1 - \gamma} \overset{①}{\le} 2\|\mathbf{W}\|_2 + \frac{\gamma^2(9 - 8\gamma)}{2(1 - \gamma)^2} \le 2.05\gamma$$

where ① uses Lemma F.10. $\qquad\square$

**Lemma\* F.16.** *If* $\|\mathbf{W}\|_2, \|\mathbf{W}^*\|_2 \le \gamma \le \frac{1}{100}$,

$$\|\mathbf{A} - [\mathbf{W}^* - \mathbf{W} + (\mathbf{W}^* - \mathbf{W})^\top - \mathrm{Diag}(\mathbf{W}^* - \mathbf{W})]\|_2 \le 9.2\gamma^2$$

*Proof.* By definition,

$$\left\| \left[ (\mathbf{I} + \mathbf{W}^*)\overline{\mathbf{I} + \mathbf{W}^*}^\top - (\mathbf{I} + \mathbf{W})\overline{\mathbf{I} + \mathbf{W}}^\top \right] - \left[ (\mathbf{W}^* - \mathbf{W}) + (\overline{\mathbf{I} + \mathbf{W}^*}^\top - \overline{\mathbf{I} + \mathbf{W}}^\top) \right] \right\|_2$$

$$= \|\mathbf{W}^*(\overline{\mathbf{I} + \mathbf{W}^*}^\top - \mathbf{I}) - \mathbf{W}(\overline{\mathbf{I} + \mathbf{W}}^\top - \mathbf{I})\|_2 \le \|\mathbf{W}^*(\overline{\mathbf{I} + \mathbf{W}^*}^\top - \mathbf{I})\|_2 + \|\mathbf{W}(\overline{\mathbf{I} + \mathbf{W}})^\top - \mathbf{I})\|_2$$

$$\le 2.05\gamma^2 + 2.05\gamma^2 = 4.1\gamma^2$$

where the last inequality uses Lemma F.15. Below we further approximate $\overline{\mathbf{I} + \mathbf{W}^*}^\top - \overline{\mathbf{I} + \mathbf{W}}^\top$.

$$\left\| \left[ \overline{\mathbf{I} + \mathbf{W}^*}^\top - \overline{\mathbf{I} + \mathbf{W}}^\top \right] - \left[ (\mathbf{W}^* - \mathbf{W})^\top - \mathrm{Diag}(\mathbf{W}^* - \mathbf{W}) \right] \right\|_2$$

$$= \left\| \mathrm{Diag}(\overline{\mathbf{I} + \mathbf{W}^*}^\top - \overline{\mathbf{I} + \mathbf{W}}^\top) + \text{Off-Diag}(\overline{\mathbf{I} + \mathbf{W}^*}^\top - \overline{\mathbf{I} + \mathbf{W}}^\top) - \left[ (\mathbf{W}^* - \mathbf{W})^\top - \mathrm{Diag}(\mathbf{W}^* - \mathbf{W}) \right] \right\|_2$$

$$\overset{①}{\le} \|\text{Off-Diag}(\overline{\mathbf{I} + \mathbf{W}^*}^\top - \overline{\mathbf{I} + \mathbf{W}}^\top) - \text{Off-Diag}(\mathbf{W}^* - \mathbf{W})^\top\|_2 + \frac{\gamma^2}{(1 - \gamma)^2}$$

$$\overset{②}{\le} \frac{4\gamma^2}{1 - \gamma} + \frac{\gamma^2}{(1 - \gamma)^2} \le 5.1\gamma^2$$

where ① uses Lemma F.15, ② uses Lemma F.15 Combining everything,

$$\|\mathbf{A} - [\mathbf{W}^* - \mathbf{W} + (\mathbf{W}^* - \mathbf{W})^\top - \mathrm{Diag}(\mathbf{W}^* - \mathbf{W})]\|_2 \le 9.2\gamma^2 \qquad \square$$

Using Lemma F.10, we immediately have the following corollary.

**Corollary F.17.** $\|\mathrm{Diag}(\mathbf{A}) - \mathrm{Diag}(\mathbf{W}^* - \mathbf{W})\|_2 \le 9.2\gamma^2$.

**Lemma\* F.18.** *For* $\eta \le \frac{1}{\pi d}$,

$$\left\| \mathbf{I} - \eta \left( \frac{\pi}{2} u u^\top + \left( \frac{\pi}{2} + 1 \right) \mathbf{I} \right) \right\|_2 \le \left( 1 - \eta \left( \frac{\pi}{2} + 1 \right) \right)$$

*Proof.* Consider another basis $(e_1', \cdots, e_d')$ where $e_1' = \frac{u}{\|u\|_2}$. For every unit vector $v = (v_1, \cdots, v_d)$ in this new space, we know

$$v^T \left( \mathbf{I} - \eta \left( \frac{\pi}{2} u u^\top + \left( \frac{\pi}{2} + 1 \right) \mathbf{I} \right) \right) v = \|v\|_2^2 - \eta \left( \frac{\pi}{2} + 1 \right) \|v\|_2^2 - \frac{\pi \eta d}{2} v_1^2$$

Hence we get

$$0 \le v^T \left( \mathbf{I} - \eta \left( \frac{\pi}{2} u u^\top + \left( \frac{\pi}{2} + 1 \right) \mathbf{I} \right) \right) v \le \left( 1 - \eta \left( \frac{\pi}{2} + 1 \right) \right) \|v\|_2^2$$

By definition of matrix norm, the lemma follows. $\qquad \square$

# G   Proofs for Section B

## G.1   Proof for Claim B.1

Comparing with Lemma 2.1, we know that for fixed $j$, $\mathbf{P}_{1,j}$ is already contained in $-\nabla \mathsf{L}(\mathbf{W})_j$ as the first term, while $\mathbf{P}_{3,j}$ is simply the summand when $i = j$, ignoring the first term. Below we show how to obtain $\mathbf{P}_{2,j}$ from $i \ne j$ cases. We will bound the approximation error in Lemma B.2 and

Lemma B.3.

$$\sum_{i \neq j} \left( \left( \frac{\pi}{2} - \theta_{i^*,j} \right)(e_i + w_i^*) - \left( \frac{\pi}{2} - \theta_{i,j} \right)(e_i + w_i) + (\|e_i + w_i^*\| \sin \theta_{i^*,j} - \|e_i + w_i\| \sin \theta_{i,j}) \overline{e_j + w_j} \right)$$

$$\approx \sum_{i \neq j} \left( \langle \overline{e_i + w_i^*}, \overline{e_j + w_j} \rangle (e_i + w_i^*) - \langle \overline{e_i + w_i}, \overline{e_j + w_j} \rangle (e_i + w_i) \right)$$

$$+ \sum_{i \neq j} \left( \|e_i + w_i^*\| \left( 1 - \frac{1}{2} \langle \overline{e_i + w_i^*}, \overline{e_j + w_j} \rangle^2 \right) - \|e_i + w_i\| \left( 1 - \frac{1}{2} \langle \overline{e_i + w_i}, \overline{e_j + w_j} \rangle^2 \right) \right) \overline{e_j + w_j}$$

$$= \sum_{i \neq j} ((e_i + w_i^*) \overline{e_i + w_i^*}^\top - (e_i + w_i) \overline{e_i + w_i}^\top) \overline{e_j + w_j}$$

$$+ \sum_{i \neq j} \left( \|e_i + w_i^*\| - \|e_i + w_i\| - \frac{1}{2} \overline{e_j + w_j}^\top \overline{e_i + w_i^*} \|e_i + w_i^*\| \overline{e_i + w_i^*}^\top \overline{e_j + w_j} \right.$$

$$+ \frac{1}{2} \overline{e_j + w_j}^\top \overline{e_i + w_i} \|e_i + w_i\| \overline{e_i + w_i}^\top \overline{e_j + w_j} \right) \overline{e_j + w_j}$$

$$= \mathbf{A}_j \overline{e_j + w_j} + \left( \sum_{i \neq j} (\|e_i + w_i^*\| - \|e_i + w_i\|) - \sum_{i \neq j} \frac{1}{2} \overline{e_j + w_j}^\top (e_i + w_i^*) \overline{e_i + w_i^*}^\top \overline{e_j + w_j} \right.$$

$$+ \sum_{i \neq j} \frac{1}{2} \overline{e_j + w_j}^\top (e_i + w_i) \overline{e_i + w_i}^\top \overline{e_j + w_j} \right) \overline{e_j + w_j}$$

$$= \mathbf{A}_j \overline{e_j + w_j} + \left( g_j - \frac{1}{2} \overline{e_j + w_j}^\top \mathbf{A}_j \overline{e_j + w_j} \right) \overline{e_j + w_j} = \mathbf{P}_{2,j}.$$

## G.2 Proof for Lemma B.2

In order to prove this lemma, we bound the approximation loss of $\theta_{i,j}, \theta_{i^*,j}$ in Lemma G.1, and the approximation loss of $\sin \theta_{i,j}, \sin \theta_{i^*,j}$ in Lemma G.2.

**Lemma* G.1** (Approximation loss related to $\theta_{i,j}, \theta_{i^*,j}$). *If* $\|\mathbf{W}\|_2, \|\mathbf{W}^*\|_2 \leq \gamma \leq \frac{1}{100}$,

$$\sum_{j=1}^d \sum_{i \neq j} \left| \left\langle \left( \frac{\pi}{2} - \theta_{i^*,j} - \langle \overline{e_i + w_i^*}, \overline{e_j + w_j} \rangle \right)(e_i + w_i^*) - \left( \frac{\pi}{2} - \theta_{i,j} - \langle \overline{e_i + w_i}, \overline{e_j + w_j} \rangle \right)(e_i + w_i), w_j^* - w_j \right\rangle \right|$$

$$\leq 0.083 \|\mathbf{W}^* - \mathbf{W}\|_F^2$$

*Proof.* By definition, $\frac{\pi}{2} - \theta_{i^*,j} = \arcsin \langle \overline{e_i + w_i^*}, \overline{e_j + w_j} \rangle$, and $\frac{\pi}{2} - \theta_{i,j} = \arcsin \langle \overline{e_i + w_i}, \overline{e_j + w_j} \rangle$.

The Taylor series of $\arcsin x$ at $x = 0$ is $\sum_{k=0}^\infty \frac{(2k)!}{4^k (k!)^2 (2k+1)} x^{2k+1}$, where for $k \geq 1$,

$$\frac{(2k)!}{4^k (k!)^2 (2k+1)} \leq \frac{1}{6} \tag{11}$$

Thus,

$$\sum_{j=1}^{d}\sum_{i\neq j}\left|\left\langle (\frac{\pi}{2}-\theta_{i^*,j}-\overline{\langle e_i+w_i^*},\overline{e_j+w_j}\rangle)(e_i+w_i^*)-(\frac{\pi}{2}-\theta_{i,j}-\overline{\langle e_i+w_i},\overline{e_j+w_j}\rangle)(e_i+w_i),w_j^*-w_j\right\rangle\right|$$

$$\overset{①}{\leq}\sum_{j=1}^{d}\sum_{i\neq j}\sum_{k=1}^{\infty}\frac{1}{6}\left|\left\langle \overline{\langle e_i+w_i^*},\overline{e_j+w_j}\rangle^{2k+1}(e_i+w_i^*)-\overline{\langle e_i+w_i},\overline{e_j+w_j}\rangle^{2k+1}(e_i+w_i),w_j^*-w_j\right\rangle\right|$$

$$\overset{②}{\leq}\sum_{j=1}^{d}\sum_{i\neq j}\sum_{k=1}^{\infty}\frac{1}{6}\left\|\overline{\langle e_i+w_i^*},\overline{e_j+w_j}\rangle^{2k+1}(e_i+w_i^*)-\overline{\langle e_i+w_i},\overline{e_j+w_j}\rangle^{2k+1}(e_i+w_i)\right\|_2\|w_j^*-w_j\|_2$$

$$\overset{③}{\leq}\sum_{j=1}^{d}\sum_{i\neq j}\sum_{k=1}^{\infty}(2.2\gamma)^{2k-2}\left(\overline{\langle e_i+w_i^*},\overline{e_j+w_j}\rangle^2+\overline{\langle e_i+w_i},\overline{e_j+w_j}\rangle^2\right)\|w_i^*-w_i\|_2\|w_j^*-w_j\|_2$$

$$\overset{④}{\leq}\sum_{j=1}^{d}\sum_{i\neq j}1.01\left(\overline{\langle e_i+w_i^*},\overline{e_j+w_j}\rangle^2+\overline{\langle e_i+w_i},\overline{e_j+w_j}\rangle^2\right)\|w_i^*-w_i\|_2\|w_j^*-w_j\|_2$$

$$\overset{⑤}{\leq}1.01\left(\sum_{j=1}^{d}\sum_{i\neq j}\left(\overline{\langle e_i+w_i^*},\overline{e_j+w_j}\rangle^2+\overline{\langle e_i+w_i},\overline{e_j+w_j}\rangle^2\right)\|w_i^*-w_i\|_2^2\right)^{\frac{1}{2}}$$

$$\left(\sum_{j=1}^{d}\sum_{i\neq j}\left(\overline{\langle e_i+w_i^*},\overline{e_j+w_j}\rangle^2+\overline{\langle e_i+w_i},\overline{e_j+w_j}\rangle^2\right)\|w_j^*-w_j\|_2^2\right)^{\frac{1}{2}}$$

$$\leq1.01\left[\sum_{i=1}^{d}\|w_i^*-w_i\|_2^2\left(\sum_{i\neq j}\left(\overline{\langle e_i+w_i^*},\overline{e_j+w_j}\rangle^2+\overline{\langle e_i+w_i},\overline{e_j+w_j}\rangle^2\right)\right)\right]^{\frac{1}{2}}$$

$$\left[\sum_{j=1}^{d}\|w_j^*-w_j\|_2^2\left(\sum_{i\neq j}\left(\overline{\langle e_i+w_i^*},\overline{e_j+w_j}\rangle^2+\overline{\langle e_i+w_i},\overline{e_j+w_j}\rangle^2\right)\right)\right]^{\frac{1}{2}}$$

$$\overset{⑥}{\leq}1.01\left(\frac{4\gamma}{(1-\gamma)^2}+\frac{4\gamma(1+\gamma)}{1-2\gamma}\right)\|\mathbf{W}^*-\mathbf{W}\|_F^2\overset{⑦}{\leq}0.083\|\mathbf{W}^*-\mathbf{W}\|_F^2$$

where ① is by Taylor series, ② uses Cauchy Schwartz, ③ uses Lemma F.7, ④ holds as $\gamma\leq\frac{1}{100}$, ⑤ uses Cauchy Schwartz, ⑥ uses Lemma F.9, ⑦ holds as $\gamma\leq\frac{1}{100}$.

□

**Lemma\* G.2** (Approximation loss related to $\sin\theta_{i,j},\sin\theta_{i^*,j}$). *If* $\|\mathbf{W}\|_2,\|\mathbf{W}^*\|_2\leq\gamma\leq\frac{1}{100}$,

$$\sum_{j=1}^{d}\sum_{i\neq j}\left|\left(\|e_i+w_i^*\|_2\left(\sin\theta_{i^*,j}-1+\frac{1}{2}\overline{\langle e_i+w_i^*},\overline{e_j+w_j}\rangle^2\right)-\right.\right.$$

$$\left.\left.\|e_i+w_i\|_2\left(\sin\theta_{i,j}-1+\frac{1}{2}\overline{\langle e_i+w_i},\overline{e_j+w_j}\rangle^2\right)\right)\overline{\langle e_j+w_j},w_j^*-w_j\rangle\right|\leq0.002\|\mathbf{W}^*-\mathbf{W}\|_F^2$$

*Proof.* By definition, we know $\theta_{i^*,j}=\arccos\overline{\langle e_i+w_i^*},\overline{e_j+w_j}\rangle$, and $\theta_{i,j}=\arccos\overline{\langle e_i+w_i},\overline{e_j+w_j}\rangle$. The Taylor series of $\sin(\arccos x)$ at $x=0$ is $1-\frac{x^2}{2}-\frac{x^4}{8}-$

$\frac{x^6}{16} - \frac{5x^8}{128} - \cdots = \sum_{k=0}^{\infty} c_k x^{2k}$, where $c_k \leq \frac{1}{8}$ for $k \geq 2$. Thus,

$$\sum_{j=1}^{d} \sum_{i \neq j} \left| \left( \|e_i + w_i^*\|_2 \left( \sin \theta_{i^*,j} - 1 + \frac{1}{2} \overline{\langle e_i + w_i^*, e_j + w_j \rangle}^2 \right) - \right. \right.$$

$$\left. \left. \|e_i + w_i\|_2 \left( \sin \theta_{i,j} - 1 + \frac{1}{2} \overline{\langle e_i + w_i, e_j + w_j \rangle}^2 \right) \right) \langle \overline{e_j + w_j}, w_j^* - w_j \rangle \right|$$

$$\overset{\textcircled{1}}{\leq} \sum_{j=1}^{d} \sum_{i \neq j} \left| \sum_{k=2}^{\infty} \frac{1}{8} \left( \|e_i + w_i\|_2 \overline{\langle e_i + w_i, e_j + w_j \rangle}^{2k} - \|e_i + w_i^*\|_2 \overline{\langle e_i + w_i^*, e_j + w_j \rangle}^{2k} \right) \right| \|w_j^* - w_j\|_2$$

$$\overset{\textcircled{2}}{\leq} \sum_{j=1}^{d} \sum_{i \neq j} \sum_{k=2}^{\infty} (2.2\gamma)^{2k-3} \left( \overline{\langle e_i + w_i, e_j + w_j \rangle}^2 + \overline{\langle e_i + w_i^*, e_j + w_j \rangle}^2 \right) \|w_i^* - w_i\|_2 \|w_j^* - w_j\|_2$$

$$\overset{\textcircled{3}}{\leq} 2.3\gamma \left( \sum_{j=1}^{d} \sum_{i \neq j} \left( \overline{\langle e_i + w_i, e_j + w_j \rangle}^2 + \overline{\langle e_i + w_i^*, e_j + w_j \rangle}^2 \right) \|w_i^* - w_i\|_2^2 \right)^{\frac{1}{2}}$$

$$\left( \sum_{j=1}^{d} \sum_{i \neq j} \left( \overline{\langle e_i + w_i, e_j + w_j \rangle}^2 + \overline{\langle e_i + w_i^*, e_j + w_j \rangle}^2 \right) \|w_j^* - w_j\|_2^2 \right)^{\frac{1}{2}}$$

$$\leq 2.3\gamma \left[ \sum_{i=1}^{d} \|w_i^* - w_i\|_2^2 \left( \sum_{j \neq i} \left( \overline{\langle e_i + w_i, e_j + w_j \rangle}^2 + \overline{\langle e_i + w_i^*, e_j + w_j \rangle}^2 \right) \right) \right]^{\frac{1}{2}}$$

$$\left[ \sum_{j=1}^{d} \|w_j^* - w_j\|_2^2 \left( \sum_{i \neq j} \left( \overline{\langle e_i + w_i, e_j + w_j \rangle}^2 + \overline{\langle e_i + w_i^*, e_j + w_j \rangle}^2 \right) \right) \right]^{\frac{1}{2}}$$

$$\overset{\textcircled{4}}{\leq} 2.3\gamma \left( \frac{4\gamma}{(1-\gamma)^2} + \frac{4\gamma(1+\gamma)}{1-2\gamma} \right) \|\mathbf{W}^* - \mathbf{W}\|_F^2 \overset{\textcircled{5}}{<} 0.002 \|\mathbf{W}^* - \mathbf{W}\|_F^2$$

where ① is by Taylor series, ② uses Lemma F.8 and Cauchy Schwartz, ③ uses Cauchy Schwartz and $\gamma \leq \frac{1}{100}$, ④ uses Lemma F.9, and ⑤ holds as $\gamma \leq \frac{1}{100}$. $\qquad \square$

*Proof for Lemma B.2.* Combining the results from Lemma G.1 and Lemma G.2, the lemma follows.
$\qquad \square$

### G.3 Proof for Lemma B.3

Denote $\boldsymbol{\Delta} \triangleq \mathbf{P} + \nabla L(\mathbf{W})$. This lemma is harder to prove than the previous one since we need to bound the spectral norm of a matrix $\boldsymbol{\Delta}$. First of all, we need to represent $\boldsymbol{\Delta}$. Again, the difference has two parts: approximation for $\theta_{i,j}, \theta_{i^*,j}$, and $\sin \theta_{i,j}, \sin \theta_{i^*,j}$. Denote the two parts as $\boldsymbol{\Delta}_1, \boldsymbol{\Delta}_2$, where $\boldsymbol{\Delta} = \boldsymbol{\Delta}_1 + \boldsymbol{\Delta}_2$. From the proof of Lemma G.1, we know the $j$-th column of the first part is

$$\boldsymbol{\Delta}_{1,j} \triangleq \sum_{i \neq j} \sum_{k=1}^{\infty} \frac{(2k)!}{4^k (k!)^2 (2k+1)} \left( \overline{\langle e_i + w_i^*, e_j + w_j \rangle}^{2k+1} (e_i + w_i^*) - \overline{\langle e_i + w_i, e_j + w_j \rangle}^{2k+1} (e_i + w_i) \right)$$

And the $j$-th column of the second part is

$$\boldsymbol{\Delta}_{2,j} \triangleq \sum_{i \neq j} \sum_{k=2}^{\infty} c_k \left( \|e_i + w_i\|_2 \overline{\langle e_i + w_i, e_j + w_j \rangle}^{2k} - \|e_i + w_i^*\|_2 \overline{\langle e_i + w_i^*, e_j + w_j \rangle}^{2k} \right) \overline{e_j + w_j}$$

Below we bound $\|\boldsymbol{\Delta}_1\|_2$ in Lemma G.3, and bounds $\|\boldsymbol{\Delta}_2\|_2$ in Lemma G.4.

**Lemma\* G.3.** *If* $\|\mathbf{W}\|_2, \|\mathbf{W}^*\|_2 \leq \gamma \leq \frac{1}{100}$, $\|\boldsymbol{\Delta}_1\|_2 \leq 3.4\gamma^2$.

*Proof.* Define $\mathbf{U}, \mathbf{V}$ such that for $i = j$, $\mathbf{U}_{i,j} = \mathbf{V}_{i,j} = 0$, and for $i \neq j$,

$$\mathbf{U}_{i,j} = \sum_{k=1}^{\infty} \frac{(2k)!}{4^k (k!)^2 (2k+1)} \langle \overline{e_i + w_i^*}, \overline{e_j + w_j} \rangle^{2k+1}, \mathbf{V}_{i,j} = \sum_{k=1}^{\infty} \frac{(2k)!}{4^k (k!)^2 (2k+1)} \langle \overline{e_i + w_i}, \overline{e_j + w_j} \rangle^{2k+1}$$

By matrix multiplication,

$$\mathbf{\Delta}_1 = \sum_{i=1}^{d} [(\mathbf{I} + \mathbf{W}^*)_{*,i} \mathbf{U}_{i,*} - (\mathbf{I} + \mathbf{W})_{*,i} \mathbf{V}_{i,*}] = (\mathbf{I} + \mathbf{W}^*)\mathbf{U} - (\mathbf{I} + \mathbf{W})\mathbf{V} \qquad (12)$$

So it suffices to bound $\|\mathbf{U}\|_2, \|\mathbf{V}\|_2$. For $i \neq j$,

$$|\mathbf{U}_{i,j}| = \left| \sum_{k=1}^{\infty} \frac{(2k)!}{4^k (k!)^2 (2k+1)} \langle \overline{e_i + w_i^*}, \overline{e_j + w_j} \rangle^{2k+1} \right| \overset{\text{①}}{\leq} \sum_{k=1}^{\infty} \frac{(2.1\gamma)^{2k-1}}{6} \langle \overline{e_i + w_i^*}, \overline{e_j + w_j} \rangle^2 \leq 0.4\gamma \langle \overline{e_i + w_i^*}, \overline{e_j + w_j} \rangle^2$$

where ① uses Lemma F.2 and (11). Now, we know

$$\|\mathbf{U}\|_1 \overset{\text{①}}{=} \max_j \sum_{i=1}^{d} |\mathbf{U}_{i,j}| \leq \max_j \sum_{i \neq j} 0.4\gamma \langle \overline{e_i + w_i^*}, \overline{e_j + w_j} \rangle^2 \overset{\text{②}}{\leq} \frac{1.6(1+\gamma)\gamma^2}{1 - 2\gamma} \leq 1.65\gamma^2$$

where ① is by definition, ② uses Lemma F.9. Similarly,

$$\|\mathbf{U}\|_\infty = \max_i \sum_{j=1}^{d} |\mathbf{U}_{i,j}| \leq \max_i \sum_{j \neq i} 0.4\gamma \langle \overline{e_i + w_i^*}, \overline{e_j + w_j} \rangle^2 \leq 1.65\gamma^2$$

By Hölder's inequality, we have

$$\|\mathbf{U}\|_2 \leq \sqrt{\|\mathbf{U}\|_1 \|\mathbf{U}\|_\infty} \leq 1.65\gamma^2$$

Now we do the same analysis for $\mathbf{V}$.

$$|\mathbf{V}_{i,j}| = \left| \sum_{k=1}^{\infty} \frac{(2k)!}{4^k (k!)^2 (2k+1)} \langle \overline{e_i + w_i}, \overline{e_j + w_j} \rangle^{2k+1} \right|$$

$$\leq \sum_{k=1}^{\infty} \frac{(2.1\gamma)^{2k-1}}{6} \langle \overline{e_i + w_i}, \overline{e_j + w_j} \rangle^2 \leq 0.4\gamma \langle \overline{e_i + w_i}, \overline{e_j + w_j} \rangle^2$$

Hence, $\|\mathbf{V}\|_1 = \max_j \sum_{i=1}^{d} |\mathbf{V}_{i,j}| \leq \max_j \sum_{i \neq j} 0.4\gamma \langle \overline{e_i + w_i}, \overline{e_j + w_j} \rangle^2 \leq 1.65\gamma^2$. Similarly, $\|\mathbf{V}\|_\infty \leq 1.65\gamma^2$, and by Hölder's inequality, $\|\mathbf{V}\|_2 \leq \sqrt{\|\mathbf{V}\|_1 \|\mathbf{V}\|_\infty} \leq 1.65\gamma^2$. Using (12), we get

$$\|\mathbf{\Delta}_1\|_2 \leq \|\mathbf{I} + \mathbf{W}^*\|_2 \|\mathbf{U}\|_2 + \|\mathbf{I} + \mathbf{W}\|_2 \|\mathbf{V}\|_2 \leq 2(1+\gamma)1.65\gamma^2 < 3.4\gamma^2 \qquad \square$$

**Lemma\* G.4.** *If* $\|\mathbf{W}\|_2, \|\mathbf{W}^*\|_2 \leq \gamma \leq \frac{1}{100}$, $\|\mathbf{\Delta}_2\|_2 \leq 6\gamma^3$.

*Proof.* By definition, we can write

$$\mathbf{\Delta}_2 = \overline{\mathbf{I} + \mathbf{W}} \mathrm{Diag} \left\{ \sum_{i \neq j} \sum_{k=2}^{\infty} c_k \left( \|e_i + w_i\|_2 \langle \overline{e_i + w_i}, \overline{e_j + w_j} \rangle^{2k} - \|e_i + w_i^*\|_2 \langle \overline{e_i + w_i^*}, \overline{e_j + w_j} \rangle^{2k} \right) \right\}_{j=1}^{d}$$

So it suffices to bound the norm of the diagonal matrix, which is the maximum of the diagonal entries. For any $j \in [d]$, we have

$$\left| \sum_{i \neq j} \sum_{k=2}^{\infty} c_k \left( \|e_i + w_i\|_2 \langle \overline{e_i + w_i}, \overline{e_j + w_j} \rangle^{2k} - \|e_i + w_i^*\|_2 \langle \overline{e_i + w_i^*}, \overline{e_j + w_j} \rangle^{2k} \right) \right|$$

$$\leq \sum_{i \neq j} \sum_{k=2}^{\infty} \frac{1}{8} \left( \|e_i + w_i\|_2 \overline{\langle e_i + w_i}, \overline{e_j + w_j} \rangle^{2k} | + \| |e_i + w_i^*\|_2 \langle \overline{e_i + w_i^*}, \overline{e_j + w_j} \rangle^{2k} \right)$$

$$\overset{①}{\leq} \sum_{i \neq j} \sum_{k=2}^{\infty} \frac{1}{4} (1 + \gamma)(2.1\gamma)^{2k-2} \left( \langle \overline{e_i + w_i}, \overline{e_j + w_j} \rangle^2 + \langle \overline{e_i + w_i^*}, \overline{e_j + w_j} \rangle^2 \right)$$

$$\overset{②}{\leq} 0.6\gamma^2 \sum_{i \neq j} \left( \langle \overline{e_i + w_i}, \overline{e_j + w_j} \rangle^2 + \langle \overline{e_i + w_i^*}, \overline{e_j + w_j} \rangle^2 \right)$$

$$\overset{③}{\leq} 0.6\gamma^2 \left( \frac{4\gamma}{(1 - \gamma)^2} + \frac{4\gamma(1 + \gamma)}{1 - 2\gamma} \right) < 5\gamma^3$$

where ① uses Lemma F.2, ② uses $\gamma \leq \frac{1}{100}$, ③ uses Lemma F.9. So we get $\|\boldsymbol{\Delta}_2\|_2 \leq \frac{1+\gamma}{1-\gamma} 5\gamma^3 \leq 6\gamma^3$. $\qquad \square$

*Proof for Lemma B.3.* Combining the results from Lemma G.3 and Lemma G.4, the lemma follows. $\qquad \square$

## H  Proofs for Section C

### H.1  Proof for Lemma C.1

In Lemma B.3, we use $\mathbf{P}(\mathbf{W})$ to approximate $-\nabla \mathsf{L}(\mathbf{W})$ in terms of spectral norm, with approximation loss $3.5\gamma^2$. Below we will get $\mathbf{Q}(\mathbf{W})$ from $\mathbf{P}(\mathbf{W})$ by removing a few more lower order terms.

By definition 2.3, we have

$$\mathbf{P}_{2,j} = g\overline{e_j + w_j} - (\|e_j + w_j^*\|_2 - \|e_j + w_j\|_2)\overline{e_j + w_j} + \left( \mathbf{I} - \frac{1}{2}\overline{e_j + w_j} \cdot \overline{e_j + w_j}^\top \right) \mathbf{A}\overline{e_j + w_j}$$

$$+ \left( \mathbf{I} - \frac{1}{2}\overline{e_j + w_j} \cdot \overline{e_j + w_j}^\top \right)(e_j + w_j) - \left( \mathbf{I} - \frac{1}{2}\overline{e_j + w_j} \cdot \overline{e_j + w_j}^\top \right)(e_j + w_j^*)\overline{e_j + w_j^*}^\top \overline{e_j + w_j}$$

$$= g\overline{e_j + w_j} - (\|e_j + w_j^*\|_2 - \|e_j + w_j\|_2)\overline{e_j + w_j} + \left( \mathbf{I} - \frac{1}{2}\overline{e_j + w_j} \cdot \overline{e_j + w_j}^\top \right) \mathbf{A}\overline{e_j + w_j}$$

$$+ \frac{1}{2}(e_j + w_j) - (e_j + w_j^*)\overline{e_j + w_j^*}^\top \overline{e_j + w_j} + \frac{1}{2}\overline{e_j + w_j}\|e_j + w_j^*\|_2 (\overline{e_j + w_j^*}^\top \overline{e_j + w_j})^2$$

$$= g\overline{e_j + w_j} + \left( \mathbf{I} - \frac{1}{2}\overline{e_j + w_j} \cdot \overline{e_j + w_j}^\top \right) \mathbf{A}\overline{e_j + w_j} + \frac{3}{2}(e_j + w_j) - \overline{e_j + w_j^*}^\top \overline{e_j + w_j}(e_j + w_j^*)$$

$$+ \left( \frac{1}{2}\|e_j + w_j^*\|_2 (\overline{e_j + w_j^*}^\top \overline{e_j + w_j})^2 - \|e_j + w_j^*\|_2 \right) \overline{e_j + w_j}$$

$$= g\overline{e_j + w_j} + \left( \mathbf{I} - \frac{1}{2}\overline{e_j + w_j} \cdot \overline{e_j + w_j}^\top \right) \mathbf{A}\overline{e_j + w_j} - w_j^* + w_j + (1 - \overline{e_j + w_j^*}^\top \overline{e_j + w_j})(e_j + w_j^*)$$

$$+ \left( \frac{1}{2}\|e_j + w_j\|_2 + \frac{1}{2}\|e_j + w_j^*\|_2 (\overline{e_j + w_j^*}^\top \overline{e_j + w_j})^2 - \|e_j + w_j^*\|_2 \right) \overline{e_j + w_j}$$

Combining every column together, we get

$$\mathbf{P}_2 = g\overline{\mathbf{I} + \mathbf{W}} + \mathbf{A}\overline{\mathbf{I} + \mathbf{W}} - \frac{1}{2}\overline{\mathbf{I} + \mathbf{W}}\mathrm{Diag}(\{\overline{e_j + w_j}^\top \mathbf{A}\overline{e_j + w_j}\}_{j=1}^d) - (\mathbf{W}^* - \mathbf{W}) + \overline{\mathbf{I} + \mathbf{W}^*}\boldsymbol{\Sigma}_1 + \overline{\mathbf{I} + \mathbf{W}}\boldsymbol{\Sigma}_2$$

where

$$\boldsymbol{\Sigma}_1 = \mathrm{Diag}(\{(\|e_j + w_j^*\|_2 - \|e_j + w_j^*\|_2 \overline{e_j + w_j}^\top \overline{e_j + w_j})\}_{j=1}^d)$$

$$\boldsymbol{\Sigma}_2 = \mathrm{Diag}(\{\frac{1}{2}\|e_j + w_j\|_2 + \frac{1}{2}\|e_j + w_j^*\|_2 (\overline{e_j + w_j^*}^\top \overline{e_j + w_j})^2 - \|e_j + w_j^*\|_2\}_{j=1}^d)$$

Using Lemma F.12, we replace $\overline{e_j + w_j}^\top \mathbf{A}\overline{e_j + w_j}$ with $e_j^\top \mathbf{A} e_j$. By Lemma F.1,

$$\left\| \mathbf{P}_2 - \left[ g\overline{\mathbf{I} + \mathbf{W}} + \mathbf{A}\overline{\mathbf{I} + \mathbf{W}} - \frac{1}{2}\overline{\mathbf{I} + \mathbf{W}}\mathrm{Diag}(\mathbf{A}) - (\mathbf{W}^* - \mathbf{W}) + \overline{\mathbf{I} + \mathbf{W}^*}\boldsymbol{\Sigma}_1 + \overline{\mathbf{I} + \mathbf{W}}\boldsymbol{\Sigma}_2 \right] \right\|_2 \leq \frac{5(1+\gamma)}{2(1-\gamma)} < 2.6\gamma^2$$

We then focus on the middle two summands in the sum.

$$\mathbf{A}\overline{\mathbf{I} + \mathbf{W}} - \frac{1}{2}\overline{\mathbf{I} + \mathbf{W}}\mathrm{Diag}(\mathbf{A}) = (\mathbf{A} - \frac{1}{2}\mathrm{Diag}(\mathbf{A})) + \mathbf{A}(\overline{\mathbf{I} + \mathbf{W}} - \mathbf{I}) - \frac{1}{2}(\overline{\mathbf{I} + \mathbf{W}} - \mathbf{I})\mathrm{Diag}(\mathbf{A})$$

By Lemma F.10, $\|\mathrm{Diag}(\mathbf{A})\|_2 \leq \|\mathbf{A}\|_2$, so

$$\left\| \left[ \mathbf{A}\overline{\mathbf{I} + \mathbf{W}} - \frac{1}{2}\overline{\mathbf{I} + \mathbf{W}}\mathrm{Diag}(\mathbf{A}) \right] - \left[ \mathbf{A} - \frac{1}{2}\mathrm{Diag}(\mathbf{A}) \right] \right\|_2 = \left\| \mathbf{A}(\overline{\mathbf{I} + \mathbf{W}} - \mathbf{I}) - \frac{1}{2}(\overline{\mathbf{I} + \mathbf{W}} - \mathbf{I})\mathrm{Diag}(\mathbf{A}) \right\|_2$$

$$\leq \|\mathbf{A}\|_2 \|\overline{\mathbf{I} + \mathbf{W}} - \mathbf{I}\|_2 + \frac{1}{2}\|\overline{\mathbf{I} + \mathbf{W}} - \mathbf{I}\|_2 \|\mathrm{Diag}(\mathbf{A})\|_2 \overset{\text{①}}{\leq} \frac{3\gamma(\gamma^2 + 3)}{1 - \gamma^2} 2.05\gamma < 18.5\gamma^2$$

where ① uses Lemma F.11 and Lemma F.15.

Moreover, by Lemma E.1 term 2, we know $\|\boldsymbol{\Sigma}_1\|_2 \leq \max_{i \in [d]}(1 + \gamma)\frac{\|w_i^* - w_i\|_2^2}{2(1-2\gamma)} \leq 2.07\gamma^2$, and in $\boldsymbol{\Sigma}_2$,

$$\left| \frac{1}{2}\|e_j + w_j^*\|_2 (\overline{e_j + w_j^*}^\top \overline{e_j + w_j})^2 - \frac{1}{2}\|e_j + w_j^*\|_2 \right| \leq \frac{1}{2}(1 + \gamma)\left| \overline{e_j + w_j^*}^\top \overline{e_j + w_j} - 1 \right|\left| \overline{e_j + w_j^*}^\top \overline{e_j + w_j} + 1 \right| \leq 2.07\gamma^2$$

so the following terms approximates $\mathbf{P}_2$ with approximation loss $(2.6 + 18.5 + 2.07 + 2.07)\gamma^2 < 25.3\gamma^2$.

$$\overline{\mathbf{I} + \mathbf{W}}(g\mathbf{I} - \boldsymbol{\Sigma}_3) + \mathbf{A} - \frac{1}{2}\mathrm{Diag}(\mathbf{A}) - (\mathbf{W}^* - \mathbf{W})$$

where $\boldsymbol{\Sigma}_3 = \mathrm{Diag}(\{\frac{1}{2}\|e_j + w_j^*\|_2 - \frac{1}{2}\|e_j + w_j\|_2\}_{j=1}^d)$.

By Lemma F.16 and Corollary F.17, we know $\|\mathbf{A} - [\mathbf{W}^* - \mathbf{W} + (\mathbf{W}^* - \mathbf{W})^\top - \mathrm{Diag}(\mathbf{W}^* - \mathbf{W})]\|_2 \leq 9.2\gamma^2$ and $\|\mathrm{Diag}(\mathbf{A}) - \mathrm{Diag}(\mathbf{W}^* - \mathbf{W})\|_2 \leq 9.2\gamma^2$. Therefore, with approximation loss of $18.4\gamma^2$, we get

$$\left\| \left[ \mathbf{A} - \frac{1}{2}\mathrm{Diag}(\mathbf{A}) \right] - \left[ \mathbf{W}^* - \mathbf{W} + (\mathbf{W}^* - \mathbf{W})^\top - \frac{3}{2}\mathrm{Diag}(\mathbf{W}^* - \mathbf{W}) \right] \right\|_2 \leq 18.4\gamma^2$$

We then approximate $\boldsymbol{\Sigma}_3$:

$$\|(\overline{\mathbf{I} + \mathbf{W}})\boldsymbol{\Sigma}_3 - (\overline{\mathbf{I} + \mathbf{W}})\frac{1}{2}\mathrm{Diag}(\mathbf{W}^* - \mathbf{W})\|_2 \leq \frac{1 + \gamma}{1 - \gamma}\left( \frac{1}{2}\max_j |\|e_j + w_j^*\|_2 - \|e_j + w_j\|_2 - w_{j,j}^* + w_{j,j}| \right) < 3.1\gamma^2$$

where the last inequality is by Lemma F.13. Moreover,

$$\|\overline{\mathbf{I} + \mathbf{W}}\left( \frac{1}{2}\mathrm{Diag}(\mathbf{W}^* - \mathbf{W}) \right) - \frac{1}{2}\mathrm{Diag}(\mathbf{W}^* - \mathbf{W})\|_2$$

$$\leq \|\overline{\mathbf{I} + \mathbf{W}} - \mathbf{I}\|_2 \left\| \frac{1}{2}\mathrm{Diag}(\mathbf{W}^* - \mathbf{W}) \right\|_2 < 2.05\gamma\left( \frac{1}{2}\max_i |w_{i,i}^* - w_{i,i}| \right) < 2.05\gamma^2$$

Putting everything together, with approximation loss of $(25.3 + 18.4 + 3.1 + 2.05)\gamma^2 = 49\gamma^2$ to $\mathbf{P}_2$, we get

$$(\mathbf{W}^* - \mathbf{W})^\top - 2\mathrm{Diag}(\mathbf{W}^* - \mathbf{W}) + g\overline{\mathbf{I} + \mathbf{W}}$$

For $\mathbf{P}_3$, using the same idea in the proof of Lemma D.3, we have

$$\mathbf{P}_3 = \frac{\pi}{2}\left( \mathbf{W}^* - \mathbf{W} \right) + \left( \overline{\mathbf{I} + \mathbf{W}} - \overline{\mathbf{I} + \mathbf{W}^*} \right)\boldsymbol{\Sigma}_4 + \overline{\mathbf{I} + \mathbf{W}}\boldsymbol{\Sigma}_5$$

Figure 11: $\Delta g$ is approximately (the summation of) the projection of $\Delta w_i$ onto $\overline{e_i + w_i}$

where $\boldsymbol{\Sigma}_4 = \mathrm{Diag}(\{\theta_{j,j^*}\|e_j+w_j^*\|_2\}_{j=1}^d)$, $\boldsymbol{\Sigma}_5 = \mathrm{Diag}(\{\|e_j+w_j^*\|_2 \sin\theta_{j,j^*} - \theta_{j,j^*}\|e_j+w_j^*\|_2\}_{j=1}^d)$.
By Taylor's Theorem, we know $\|\boldsymbol{\Sigma}_5\|_2 \le \|\mathrm{Diag}(\{\|e_j + w_j^*\|_2\theta_{j,j^*}^3/3\}_{j=1}^d)\|_2$.

Notice that $\theta_{j,j^*} \le 2.002\gamma$ by Lemma E.1 term 3, and $\|\overline{\mathbf{I}+\mathbf{W}} - \overline{\mathbf{I}+\mathbf{W}^*}\|_2 \le \frac{1+\gamma}{1-\gamma} - \frac{1-\gamma}{1+\gamma} \le 4.001\gamma$.
Consequently,

$$\left\|\mathbf{P}_3 - \frac{\pi}{2}(\mathbf{W}^* - \mathbf{W})\right\|_2 \le \|\left(\overline{\mathbf{I}+\mathbf{W}} - \overline{\mathbf{I}+\mathbf{W}^*}\right)\boldsymbol{\Sigma}_4\|_2 + \|\overline{\mathbf{I}+\mathbf{W}}\boldsymbol{\Sigma}_5\|_2$$

$$< 4.001 * 2.002(1+\gamma)\gamma^2 + \frac{(1+\gamma)^2}{3(1-\gamma)}(2.002\gamma)^3 < 8.1\gamma^2 + 2.8\gamma^3 < 8.2\gamma^2$$

we only need to keep the term $\frac{\pi}{2}(\mathbf{W}^* - \mathbf{W})$ with approximation loss $8.2\gamma^2$ to $\mathbf{P}_3$.

Now, combining the approximations to $\mathbf{P}_2$ and $\mathbf{P}_3$, and Lemma B.3, we have the following matrix with $(49 + 8.2 + 3.5)\gamma^2 < 61\gamma^2$ approximation loss to $-\nabla\mathsf{L}(\mathbf{W})$:

$$\frac{\pi}{2}(\mathbf{W}^* - \mathbf{W})\left(\mathbf{I} + uu^\top\right) + (\mathbf{W}^* - \mathbf{W})^\top - 2\mathrm{Diag}(\mathbf{W}^* - \mathbf{W}) + g\overline{\mathbf{I} + \mathbf{W}}$$

where $u$ is the all 1 vector.

## H.2 Proof for Lemma C.2

By Lemma F.4, we know $|g| \le 2d\gamma$. Using Lemma C.1,

$$\|\nabla\mathsf{L}(\mathbf{W})\|_2 \le 61\gamma^2 + \left\|\frac{\pi}{2}(\mathbf{W}^* - \mathbf{W})\left(\mathbf{I} + uu^\top\right) + (\mathbf{W}^* - \mathbf{W})^\top - 2\mathrm{Diag}(\mathbf{W}^* - \mathbf{W}) + g\overline{\mathbf{I} + \mathbf{W}}\right\|_2$$

$$\le 61\gamma^2 + (d+1)\pi\gamma + 2\gamma + 4\gamma + |g|\frac{1+\gamma}{1-\gamma} < 61\gamma^2 + (d+3)\pi\gamma + 2.05d\gamma < 6d\gamma.$$

## H.3 Proof for Lemma C.3

In this proof, we use $w_j$ to represent the $j$-th column of $\mathbf{W}_t$, and denote $\triangle w_j$ as the $j$-th column of $\mathbf{G}_t$.

### H.3.1 $\Delta g_t \approx \eta\langle\mathsf{L}(\mathbf{W}_t), \overline{\mathbf{I}+\mathbf{W}_t}\rangle$

For the intuition of this section, see Figure 11. The changes in potential function $g$ is essentially the changes in $\|e_i + w_i\|_2$ (summing over $i$), which is approximately $\Delta w_i$ projected onto $\overline{e_i + w_i}$. If we write it in matrix form, we get $\Delta g_t \approx \eta\langle\mathsf{L}(\mathbf{W}_t), \overline{\mathbf{I}+\mathbf{W}_t}\rangle$.

By definition we know $\|\mathbf{G}_t\|_2 = \|\nabla\mathsf{L}(\mathbf{W}_t) + \mathbf{E}_t\|_2 \overset{①}{\le} \|\nabla\mathsf{L}(\mathbf{W}_t)\|_2 + \|\mathbf{E}_t\|_2 \overset{②}{\le} 6d\gamma + \varepsilon = G_2$, where ① uses triangle inequality, ② uses Lemma C.2. We have

$$\eta\|\triangle w_j\|_2 \le \eta\|\mathbf{G}_t\|_2 \le \frac{\gamma^2}{G_2} \le \frac{\gamma}{6d}, \qquad \eta^2\|\triangle w_j\|_2 \le \eta\|\mathbf{G}_t\|_2^2 \le \gamma^2 \qquad (13)$$

By Definition 2.2, we know

$$\triangle g_t \triangleq g_{t+1} - g_t = \sum_{j=1}^{d} \left( \frac{\langle e_j + w_j, e_j + w_j \rangle}{\|e_j + w_j\|_2} - \frac{\langle e_j + w_j - \eta \triangle w_j, e_j + w_j - \eta \triangle w_j \rangle}{\|e_j + w_j - \eta \triangle w_j\|_2} \right)$$

$$= \sum_{j=1}^{d} \left( \frac{\langle e_j + w_j, e_j + w_j \rangle \|e_j + w_j - \eta \triangle w_j\|_2 - \langle e_j + w_j - \eta \triangle w_j, e_j + w_j - \eta \triangle w_j \rangle \|e_j + w_j\|_2}{\|e_j + w_j\|_2 \|e_j + w_j - \eta \triangle w_j\|_2} \right)$$

$$= \sum_{j=1}^{d} \left( \frac{\|e_j + w_j\|_2 (\|e_j + w_j - \eta \triangle w_j\|_2 - \|e_j + w_j\|_2) + 2\eta \langle \triangle w_j, e_j + w_j \rangle - \eta^2 \|\triangle w_j\|_2^2}{\|e_j + w_j - \eta \triangle w_j\|_2} \right)$$

If we project $\eta \triangle w_j$ onto the $\overline{e_j + w_j}$ direction, we get

$$\|e_j + w_j - \eta \triangle w_j\|_2 = \sqrt{(\|e_j + w_j\|_2 - \langle \overline{e_j + w_j}, \eta \triangle w_j \rangle)^2 + (\|\eta \triangle_j\|_2^2 - \langle \overline{e_j + w_j}, \eta \triangle w_j \rangle^2)^2}$$

$$\leq \sqrt{(\|e_j + w_j\|_2 - \langle \overline{e_j + w_j}, \eta \triangle w_j \rangle)^2 + \|\eta \triangle w_j\|_2^2} \overset{①}{\leq} \|e_j + w_j\|_2 - \langle \overline{e_j + w_j}, \eta \triangle w_j \rangle + \|\eta \triangle w_j\|_2^2$$

Using (13), we have $\|e_j + w_j\|_2 - \langle \overline{e_j + w_j}, \eta \triangle w_j \rangle \geq \frac{1}{2}$. By taking square on both sides, we know ① holds. It is trivial to show that $\|e_j + w_j - \eta \triangle w_j\|_2 \geq \|e_j + w_j\|_2 - \langle \overline{e_j + w_j}, \eta \triangle w_j \rangle$, so we know

$$-\langle \overline{e_j + w_j}, \eta \triangle w_j \rangle \leq \|e_j + w_j - \eta \triangle w_j\|_2 - \|e_j + w_j\|_2 \leq -\langle \overline{e_j + w_j}, \eta \triangle w_j \rangle + \|\eta \triangle w_j\|_2^2 \tag{14}$$

Thus, with approximation loss $\sum_{j=1}^{d} \frac{\|e_j + w_j\|_2 \|\eta \triangle w_j\|_2^2}{\|e_j + w_j - \eta \triangle w_j\|_2}$, we have :

$$\triangle g_t \approx \sum_{j=1}^{d} \left( \frac{-\|e_j + w_j\|_2 \langle \overline{e_j + w_j}, \eta \triangle w_j \rangle + 2\eta \langle \triangle w_j, e_j + w_j \rangle - \eta^2 \|\triangle w_j\|_2^2}{\|e_j + w_j - \eta \triangle w_j\|_2} \right)$$

$$= \sum_{j=1}^{d} \frac{\eta \langle \triangle w_j, e_j + w_j \rangle - \eta^2 \|\triangle w_j\|_2^2}{\|e_j + w_j - \eta \triangle w_j\|_2}$$

$$= \sum_{j=1}^{d} \frac{-\eta^2 \|\triangle w_j\|_2^2}{\|e_j + w_j - \eta \triangle w_j\|_2} + \sum_{j=1}^{d} \frac{(\|e_j + w_j\|_2 - \|e_j + w_j - \eta \triangle w_j\|_2) \eta \langle \triangle w_j, \overline{e_j + w_j} \rangle}{\|e_j + w_j - \eta \triangle w_j\|_2} + \eta \langle \mathbf{G}_t, \overline{\mathbf{I} + \mathbf{W}_t} \rangle$$

Thus we get the following approximation for $\triangle g_t$.

$$|\triangle g_t - \eta \langle \mathbf{G}_t, \overline{\mathbf{I} + \mathbf{W}_t} \rangle|$$

$$\leq \sum_{j=1}^{d} \left| \frac{-\eta^2 \|\triangle w_j\|_2^2}{\|e_j + w_j - \eta \triangle w_j\|_2} + \frac{(\|e_j + w_j\|_2 - \|e_j + w_j - \eta \triangle w_j\|_2) \eta \langle \triangle w_j, \overline{e_j + w_j} \rangle}{\|e_j + w_j - \eta \triangle w_j\|_2} + \frac{\|e_j + w_j\|_2 \|\eta \triangle w_j\|_2^2}{\|e_j + w_j - \eta \triangle w_j\|_2} \right|$$

$$\overset{①}{\leq} \sum_{j=1}^{d} \left[ \left| \frac{\eta \langle \triangle w_j, \overline{e_j + w_j} \rangle (\eta \langle \triangle w_j, \overline{e_j + w_j} \rangle + \|\eta \triangle w_j\|_2^2)}{\|e_j + w_j - \eta \triangle w_j\|_2} \right| + 0.02 \eta^2 \|\triangle w_j\|_2^2 \right]$$

$$\overset{②}{\leq} \sum_{j=1}^{d} \left[ \frac{\eta^2 \|\triangle w_j\|_2^2 + \eta^3 \|\triangle w_j\|_2^3}{\|e_j + w_j - \eta \triangle w_j\|_2} + 0.02 \eta \gamma^2 \right] \overset{③}{\leq} 1.04 \eta d \gamma^2$$

where ① uses (14) again, and ② ③ uses (13), $\gamma \leq \frac{1}{100}$ and $\|e_j + w_j - \eta \triangle w_j\|_2 \geq 0.98$.

Thus $|\triangle g_t - \eta \langle \nabla \mathsf{L}(\mathbf{W}_t), \overline{\mathbf{I} + \mathbf{W}_t} \rangle| \leq 1.04 \eta d \gamma^2 + |\eta \langle \mathbf{E}_t, \overline{\mathbf{I} + \mathbf{W}_t} \rangle| < 1.04 \eta d \gamma^2 + 1.03 \eta \sqrt{d} \varepsilon$

**H.3.2** $\quad \triangle g_t \approx \eta \text{Tr}(\nabla \mathsf{L}(\mathbf{W}_t))$

We want to approximate $\overline{\mathbf{I} + \mathbf{W}_t}$ with $\mathbf{I}$. Below is the error bound.

$$|\langle \nabla \mathsf{L}(\mathbf{W}_t), \overline{\mathbf{I} + \mathbf{W}_t} - \mathbf{I}\rangle| = |\langle \nabla \mathsf{L}(\mathbf{W}_t) + \mathbf{Q}_t - \mathbf{Q}_t, \overline{\mathbf{I} + \mathbf{W}_t} - \mathbf{I}\rangle|$$

$$\overset{①}{=} d \cdot 61\gamma^2 \cdot 2.05\gamma + \sum_{i=1}^{d} 2.05\gamma \left\| (\mathbf{Q}_t - \frac{\pi}{2}(\mathbf{W}^* - \mathbf{W}_t)uu^\top)_i \right\|_2 + \left\langle \frac{\pi}{2}(\mathbf{W}^* - \mathbf{W}_t)uu^\top, \overline{\mathbf{I} + \mathbf{W}_t} - \mathbf{I}\right\rangle$$

$$\overset{②}{\leq} 1.251d\gamma^2 + 2.05d\gamma\left(\pi\gamma + 2\gamma + 4\gamma + \frac{1+\gamma}{1-\gamma}|g_t|\right) + \text{Tr}\left(\left[\frac{\pi}{2}(\mathbf{W}^* - \mathbf{W}_t)u\right]\left[u^\top\overline{\mathbf{I} + \mathbf{W}_t} - \mathbf{I}\right]^\top\right)$$

$$\overset{③}{\leq} 20d\gamma^2 + 2.1d\gamma|g_t| + \left\|\frac{\pi}{2}(\mathbf{W}^* - \mathbf{W}_t)u\right\|_2 \left\|(\overline{\mathbf{I} + \mathbf{W}_t} - \mathbf{I})u\right\|_2 \overset{④}{\leq} 20d\gamma^2 + 2.1d\gamma|g_t| + \frac{2.05\pi}{2}\|s\|_2\gamma\sqrt{d}$$

where ① uses Cauchy Schwartz and Lemma F.15, ② uses the definition of $\mathbf{Q}$ and Lemma F.1, ③ holds as for any vector $u, v$, $\text{Tr}(uv^\top) \leq \|u\|_2\|v\|_2$, ④ uses Lemma F.15.

Hence,

$$|\triangle g_t - \eta\langle \nabla \mathsf{L}(\mathbf{W}_t), \mathbf{I}\rangle|$$

$$\leq 1.04\eta d\gamma^2 + 1.03\eta\sqrt{d}\varepsilon + |\eta\langle \nabla \mathsf{L}(\mathbf{W}_t), \overline{\mathbf{I} + \mathbf{W}_t} - \mathbf{I}\rangle|$$

$$< 1.04\eta d\gamma^2 + 1.03\eta\sqrt{d}\varepsilon + 20\eta d\gamma^2 + 2.1\eta d\gamma|g_t| + \frac{2.05\pi}{2}\eta\|s\|_2\gamma\sqrt{d}$$

$$< 21.1\eta d\gamma^2 + 1.03\eta\sqrt{d}\varepsilon + 2.1\eta d\gamma|g_t| + \frac{2.05\pi}{2}\eta\|s\|_2\gamma\sqrt{d}$$

So with approximation loss of $21.1\eta d\gamma^2 + 1.03\eta\sqrt{d}\varepsilon + 2.1\eta d\gamma|g_t| + \frac{2.05\pi}{2}\eta\|s\|_2\gamma\sqrt{d}$, it suffices to consider $\eta\text{Tr}(\nabla \mathsf{L}(\mathbf{W}_t))$.

**H.3.3** $\quad \triangle g_t \approx -\eta(d + \frac{\pi}{2} - 1)g_t$

According to Lemma C.1, with approximation loss of $61\gamma^2$, we can use $-\mathbf{Q}_t$ to approximate $\nabla \mathsf{L}(\mathbf{W}_t)$.

$$\text{Tr}(\mathbf{Q}_t) = \frac{\pi}{2}\text{Tr}\left((\mathbf{W}^* - \mathbf{W}_t)\left(\mathbf{I} + uu^\top\right)\right) + \text{Tr}(\mathbf{W}^* - \mathbf{W}_t)^\top - 2\text{Tr}(\text{Diag}(\mathbf{W}^* - \mathbf{W}_t)) + g\text{Tr}(\overline{\mathbf{I} + \mathbf{W}_t})$$

$$= \left(\frac{\pi}{2} - 1\right)\text{Tr}(\mathbf{W}^* - \mathbf{W}_t) + \frac{\pi}{2}\text{Tr}\left((\mathbf{W}^* - \mathbf{W}_t)\left(uu^\top\right)\right) + g\text{Tr}(\overline{\mathbf{I} + \mathbf{W}_t})$$

$$= \left(\frac{\pi}{2} - 1\right)(\text{Tr}(\mathbf{W}^* - \mathbf{W}_t) - g_t) + \left(\frac{\pi}{2} - 1\right)g_t + \frac{\pi}{2}\text{Tr}\left((\mathbf{W}^* - \mathbf{W}_t)\left(uu^\top\right)\right) + g_t\text{Tr}(\overline{\mathbf{I} + \mathbf{W}_t})$$

Therefore,

$$\left|\text{Tr}(\mathbf{Q}_t) - g_t\text{Tr}(\mathbf{I}) - \left(\frac{\pi}{2} - 1\right)g_t\right| = \left|\text{Tr}(\mathbf{Q}_t) - \left(d + \frac{\pi}{2} - 1\right)g_t\right|$$

$$\leq \left|\left(\frac{\pi}{2} - 1\right)(\text{Tr}(\mathbf{W}^* - \mathbf{W}_t) - g_t) + \frac{\pi}{2}\text{Tr}\left((\mathbf{W}^* - \mathbf{W}_t)\left(uu^\top\right)\right) + g_t(\text{Tr}(\overline{\mathbf{I} + \mathbf{W}_t} - \mathbf{I}))\right|$$

$$\overset{①}{\leq} 6.07\left(\frac{\pi}{2} - 1\right)d\gamma^2 + \frac{\pi}{2}\|s_t\|_2\sqrt{d} + 2.05|g_t|d\gamma$$

where ① uses Lemma F.14 and Lemma F.15. Thus,

$$\left|\triangle g_t - \left[-\eta\left(d + \frac{\pi}{2} - 1\right)g_t\right]\right|$$

$$\leq \eta\left[21.1d\gamma^2 + 1.03\sqrt{d}\varepsilon + 2.1d\gamma|g_t| + \frac{2.05\pi}{2}\|s\|_2\gamma\sqrt{d} + 61d\gamma^2 + 2.05|g_t|d\gamma + 6.07\left(\frac{\pi}{2} - 1\right)d\gamma^2 + \frac{\pi}{2}\|s_t\|_2\sqrt{d}\right]$$

$$\leq \eta\left[86d\gamma^2 + 1.03\sqrt{d}\varepsilon + 4.15d\gamma|g_t| + 4.8\|s_t\|_2\gamma\sqrt{d}\right]$$

Now we have

$$|g_{t+1}| = |g_t + \triangle g_t| \leq \left(1 - \eta\left(d + \frac{\pi}{2} - 1 - 4.15d\gamma\right)\right)|g_t| + 86\eta d\gamma^2 + 1.03\eta\sqrt{d}\varepsilon + 4.8\eta\|s_t\|_2\gamma\sqrt{d}$$

$$\leq (1 - 0.95\eta d)|g_t| + 86\eta d\gamma^2 + 1.03\eta\sqrt{d}\varepsilon + 4.8\eta\|s_t\|_2\gamma\sqrt{d}$$

## H.4 Proof for Lemma C.4

By definition of $s_t$,

$$\triangle s_t \triangleq s_{t+1} - s_t = (\mathbf{W}_t - \mathbf{W}_{t+1})u = \eta(\nabla\mathsf{L}(\mathbf{W}_t) + \mathbf{E}_t)u = -\eta\mathbf{Q}_t u + \eta(\mathbf{Q}_t + \nabla\mathsf{L}(\mathbf{W}_t) + \mathbf{E}_t)u$$

By definition of $\mathbf{Q}_t$,

$$\mathbf{Q}_t u = \left(\frac{\pi}{2}(\mathbf{W}^* - \mathbf{W}_t)\left(\mathbf{I} + uu^\top\right) + (\mathbf{W}^* - \mathbf{W}_t)^\top - 2\mathrm{Diag}(\mathbf{W}^* - \mathbf{W}_t) + g_t\overline{\mathbf{I} + \mathbf{W}_t}\right)u$$

$$=\frac{(d+1)\pi}{2}s_t + \left((\mathbf{W}^* - \mathbf{W}_t)^\top - 2\mathrm{Diag}(\mathbf{W}^* - \mathbf{W}_t) + g_t\overline{\mathbf{I} + \mathbf{W}_t}\right)u$$

Thus, we know

$$\left\|\mathbf{Q}_t u - \frac{(d+1)\pi}{2}s_t\right\|_2 = \left\|\left((\mathbf{W}^* - \mathbf{W}_t)^\top - 2\mathrm{Diag}(\mathbf{W}^* - \mathbf{W}_t) + g_t\overline{\mathbf{I} + \mathbf{W}_t}\right)u\right\|_2$$

$$\leq\sqrt{d}\left(\|(\mathbf{W}^* - \mathbf{W}_t)^\top\|_2 + 2\|\mathrm{Diag}(\mathbf{W}^* - \mathbf{W}_t)\|_2 + \|g_t\overline{\mathbf{I} + \mathbf{W}_t}\|_2\right)$$

$$\overset{①}{\leq}\sqrt{d}\left(2\gamma + 4\gamma + |g_t|\frac{1+\gamma}{1-\gamma}\right) < (6\gamma + 1.03|g_t|)\sqrt{d}$$

where ① uses Lemma F.1 and Lemma F.10.

By Lemma C.1, $\|\triangle s_t - [-\eta\frac{(d+1)\pi}{2}s_t]\|_2 < \eta(6\gamma + 1.03|g_t|)\sqrt{d} + \eta\|(\mathbf{Q}_t + \nabla\mathsf{L}(\mathbf{W}_t) + \mathbf{E}_t)u\|_2 \leq \eta(6.61\gamma + 1.03|g_t| + \varepsilon)\sqrt{d}$.

## H.5 Proof for Lemma C.5

Combining Lemma C.3 and Lemma C.4, we get

$$|g_{t+1}| + \|s_{t+1}\|_2$$

$$\leq(1 - 0.95\eta d)(|g_t| + \|s_t\|_2) + \eta(6.6\gamma + 1.03|g_t| + \varepsilon)\sqrt{d} + 86\eta d\gamma^2 + 1.03\eta\sqrt{d}\varepsilon + (4.8\eta\gamma\sqrt{d} - 0.62\eta d)\|s_t\|_2$$

$$\overset{①}{\leq}(1 - 0.95\eta d)(|g_t| + \|s_t\|_2) + 6.6\eta\gamma\sqrt{d} + 86\eta d\gamma^2 + \eta 1.03|g_t|\sqrt{d} + 2.03\eta\sqrt{d}\varepsilon$$

$$\overset{②}{\leq}(1 - 0.84\eta d)(|g_t| + \|s_t\|_2) + 6.6\eta\gamma\sqrt{d} + 87\eta d\gamma^2$$

where ① uses $\gamma \leq \frac{1}{100}$, $d \geq 100$, ② uses $\varepsilon \leq \gamma^2$ and $d \geq 100$. So if the following inequality holds, $|g_t| + \|s_t\|_2$ will always decrease by factor at least $1 - 0.5\eta d$.

$$0.34\eta d(|g_t| + \|s_t\|_2) \geq 6.6\eta\gamma\sqrt{d} + 87\eta d\gamma^2$$

Which gives

$$|g_t| + \|s_t\|_2 \geq \frac{6.6\eta\gamma\sqrt{d} + 87\eta d\gamma^2}{0.34\eta d} = \frac{6.6\gamma}{0.34\sqrt{d}} + \frac{87\gamma^2}{0.34}$$

where the last expression is smaller than $4.5\gamma$. Hence, $|g_t| + \|s_t\|_2$ will keep decreasing by $1 - 0.5\eta d$ as long as it is larger than $4.5\gamma$. So we have $\|s_t\|_2 \leq 4.5\gamma$. Now plug it back to the updating rule of $|g_t|$:

$$|g_{t+1}| \leq(1 - 0.95\eta d)|g_t| + 86\eta d\gamma^2 + 1.03\eta\sqrt{d}\varepsilon + 4.8\eta\|s_t\|_2\gamma\sqrt{d}$$

$$\leq(1 - 0.95\eta d)|g_t| + 86\eta d\gamma^2 + 1.03\eta\sqrt{d}\varepsilon + 21.6\eta\gamma^2\sqrt{d}$$

In order to get factor $1 - 0.5\eta d$, we have

$$0.45\eta d|g_t| \geq 86\eta d\gamma^2 + 1.03\eta\sqrt{d}\varepsilon + 21.6\eta\gamma^2\sqrt{d}$$

Solve this inequality, we get

$$\frac{86\eta d\gamma^2 + 1.03\eta\sqrt{d}\varepsilon + 21.6\eta\gamma^2\sqrt{d}}{0.45\eta d} = \frac{86\gamma^2}{0.45} + \frac{1.03\varepsilon + 21.6\gamma^2}{0.45\sqrt{d}} \leq 197\gamma^2$$

The last inequality uses $d \geq 100, \varepsilon \leq \gamma^2$. So even after $|g_t| + \|s_t\|_2$ is below $4.5\gamma$, $|g_t|$ will keep decreasing by factor $1 - 0.5\eta d$ until it is smaller than $197\gamma^2$.

Finally we bound the number of steps to arrive $197\gamma^2$. Let $\gamma = \frac{1}{400}, \gamma_0 = \frac{1}{8000}$. Again, the constants here are pretty loose. Since $|g_t| \leq (1-0.5\eta d)^t|g_0| \leq (1-0.5\eta d)^t 2d\gamma_0$, in order to let $g_t \leq 197\gamma^2$, it suffices to have $t \geq \frac{\log \frac{197\gamma^2}{2d\gamma_0}}{\log(1 - \frac{\eta d}{2})}$. Since $\eta d$ is small, by Taylor expansion we know $\log(1 - \frac{\eta d}{2}) \approx -\frac{\eta d}{2}$. Thus, it suffices to let $t \geq \frac{2\log(0.203d)}{\eta d}$. Notice that $\frac{\log(0.203d)}{d}$ is decreasing for $d \geq 100$, we know it suffices to let $t \geq \frac{1}{16\eta}$.

## H.6 Proof for Lemma C.6

Let $\mathbf{H} = \mathbf{W} - \mathbf{W}^*$, by the updating rule of $\mathbf{W}_t$ and the definition of $\mathbf{Q}_t$, we know
$$\mathbf{H}_{t+1} = \mathbf{H}_t - \eta\mathbf{H}_t\left(\frac{\pi}{2}uu^\top + \frac{\pi}{2}\right) - \eta\mathbf{H}_t^\top + 2\eta\mathrm{Diag}(\mathbf{H}_t) + \eta g_t\overline{\mathbf{I} + \mathbf{W}} - \eta(\mathbf{G}_t + \mathbf{Q}_t)$$
That gives,
$$\|\mathbf{H}_{t+1} + \mathbf{H}_{t+1}^\top\|_2$$
$$\leq \left\|(\mathbf{H}_t + \mathbf{H}_t^\top)\left(\mathbf{I} - \eta\left(\frac{\pi}{2}uu^\top + \frac{\pi}{2} + 1\right)\right)\right\|_2 + 2\eta\left\|\mathrm{Diag}(\mathbf{H}_t + \mathbf{H}_t^\top)\right\|_2 + 2\eta|g_t|\|\overline{\mathbf{I} + \mathbf{W}}\|_2 + 2\eta\|\mathbf{E}_t + \nabla\mathsf{L}(\mathbf{W}_t) + \mathbf{Q}_t\|_2$$
$$\overset{①}{\leq} \left(\mathbf{I} - \eta\left(\frac{\pi}{2} + 1\right)\right)\|\mathbf{H}_t + \mathbf{H}_t^\top\|_2 + 2\eta\|\mathbf{H}_t + \mathbf{H}_t^\top\|_2 + \frac{2(1+\gamma)\eta|g_t|}{1 - \gamma} + 2\eta\varepsilon + 122\eta\gamma^2$$
$$\overset{②}{\leq} \left(\mathbf{I} - \eta\left(\frac{\pi}{2} - 1\right)\right)\|\mathbf{H}_t + \mathbf{H}_t^\top\|_2 + 2.05\eta|g_t| + 124\eta\gamma^2 \tag{15}$$
where ① uses Lemma F.18, Lemma F.10, $\|\mathbf{E}_t\|_2 \leq \varepsilon$ and Lemma C.1. ② uses $\varepsilon \leq \gamma^2$ and $\gamma \leq \frac{1}{100}$.

Similarly, we get
$$\|\mathbf{H}_{t+1} - \mathbf{H}_{t+1}^\top\|_2$$
$$\overset{①}{\leq} \left\|(\mathbf{H}_t - \mathbf{H}_t^\top)\left(\mathbf{I} - \eta\left(\frac{\pi}{2}uu^\top + \frac{\pi}{2} - 1\right)\right)\right\|_2 + \eta|g_t|\|\overline{\mathbf{I} + \mathbf{W}} - \mathbf{I} + \mathbf{I} - \overline{\mathbf{I} + \mathbf{W}}^\top\|_2 + 2\eta\|\mathbf{E}_t + \nabla\mathsf{L}(\mathbf{W}_t) + \mathbf{Q}_t\|_2$$
$$\overset{②}{\leq} \left(\mathbf{I} - \eta\left(\frac{\pi}{2} - 1\right)\right)\|\mathbf{H}_t - \mathbf{H}_t^\top\|_2 + 4.10\eta\gamma|g_t| + 124\eta\gamma^2 \tag{16}$$
where ① holds as the diagonal terms cancel out, ② uses Lemma F.18, Lemma F.15.

Adding (15) and (16), we get
$$\|\mathbf{H}_{t+1} + \mathbf{H}_{t+1}^\top\|_2 + \|\mathbf{H}_{t+1} - \mathbf{H}_{t+1}^\top\|_2$$
$$\leq \left(\mathbf{I} - \eta\left(\frac{\pi}{2} - 1\right)\right)\left(\|\mathbf{H}_t + \mathbf{H}_t^\top\|_2 + \|\mathbf{H}_t - \mathbf{H}_t^\top\|_2\right) + 2.1\eta|g_t| + 248\eta\gamma^2 \tag{17}$$

For any $T > 0$, by applying (17) recursively, we have
$$\|\mathbf{H}_T + \mathbf{H}_T^\top\|_2 + \|\mathbf{H}_T - \mathbf{H}_T^\top\|_2 \leq \|\mathbf{H}_0 + \mathbf{H}_0^\top\|_2 + \|\mathbf{H}_0 - \mathbf{H}_0^\top\|_2 + 2.1\eta\sum_{t=0}^{T-1}|g_t| + 248\eta T\gamma^2$$

By Lemma F.4 we know $|g_0| \leq 2d\gamma_0$, so $2.1\eta\sum_{t=0}^{T-1}|g_t| \leq \frac{2.1\eta|g_0|(1-(1-0.5\eta d)^T)}{(0.5\eta d)} \leq \frac{4.2|g_0|}{d} \leq 8.4\gamma_0$.

By the proof of Lemma C.5, we know $T \leq \frac{1}{16\eta}$, so $248\eta T\gamma^2 \leq 15.5\gamma^2$.

By triangle inequality, we know $\|\mathbf{H}_0\|_2 \leq \|\mathbf{W}_0\|_2 + \|\mathbf{W}^*\|_2 \leq 2\gamma_0$, so $\|\mathbf{H}_0 + \mathbf{H}_0^\top\|_2 + \|\mathbf{H}_0 - \mathbf{H}_0^\top\|_2 \leq 4\|\mathbf{H}_0\|_2 \leq 8\gamma_0$.

By triangle inequality again we get
$$\|\mathbf{H}_T\|_2 \leq \|\mathbf{H}_T + \mathbf{H}_T^\top\|_2 + \|\mathbf{H}_T - \mathbf{H}_T^\top\|_2 \leq \|\mathbf{H}_0 + \mathbf{H}_0^\top\|_2 + \|\mathbf{H}_0 - \mathbf{H}_0^\top\|_2 + 19\gamma^2 + 8.4\gamma_0 \leq 16.4\gamma_0 + 15.5\gamma^2$$

Recall we set $\gamma = \frac{1}{400}, \gamma_0 = \frac{1}{8000}$ in the proof of Lemma C.5, we know $\|\mathbf{W}_T\|_2 \leq \|\mathbf{W}^*\|_2 + \|\mathbf{H}_T\|_2 \leq 17.4\gamma_0 + 15.5\gamma^2 \leq \frac{1}{440} \leq \gamma$.

## H.7 Proof for Lemma C.7

First, by the proof of Lemma C.5, we know $|g_t|$ will keep small if $\|\mathbf{W}_t\|_2 \leq \gamma \leq \frac{1}{100}$.

Adding (15) and (16), we get

$$
\begin{aligned}
&\|\mathbf{H}_{t+1} + \mathbf{H}_{t+1}^\top\|_2 + \|\mathbf{H}_{t+1} - \mathbf{H}_{t+1}^\top\|_2 \\
&\leq \left(\mathbf{I} - \eta\left(\frac{\pi}{2} - 1\right)\right)\left(\|\mathbf{H}_{t+1} + \mathbf{H}_{t+1}^\top\|_2 + \|\mathbf{H}_{t+1} - \mathbf{H}_{t+1}^\top\|_2\right) + 2.1\eta|g_t| + 248\eta\gamma^2 \\
&\overset{①}{\leq} \left(\mathbf{I} - \eta\left(\frac{\pi}{2} - 1\right)\right)\left(\|\mathbf{H}_{t+1} + \mathbf{H}_{t+1}^\top\|_2 + \|\mathbf{H}_{t+1} - \mathbf{H}_{t+1}^\top\|_2\right) + 661\eta\gamma^2
\end{aligned}
\tag{18}
$$

where ① holds as $|g_t| \leq 197\gamma^2$. So either $\|\mathbf{H}_{t+1} + \mathbf{H}_{t+1}^\top\|_2 + \|\mathbf{H}_{t+1} - \mathbf{H}_{t+1}^\top\|_2$ keeps decreasing, or it increases, i.e.,

$$
\eta\left(\frac{\pi}{2} - 1\right)\left(\|\mathbf{H}_{t+1} + \mathbf{H}_{t+1}^\top\|_2 + \|\mathbf{H}_{t+1} - \mathbf{H}_{t+1}^\top\|_2\right) \leq 197\eta\gamma^2
$$

That gives,

$$
\|\mathbf{H}_{t+1} + \mathbf{H}_{t+1}^\top\|_2 + \|\mathbf{H}_{t+1} - \mathbf{H}_{t+1}^\top\|_2 \leq \frac{197\gamma^2}{\frac{\pi}{2} - 1} \leq 346\gamma^2
$$

Therefore, combined with the proof of Lemma C.6, we know $\|\mathbf{H}_{t+1} + \mathbf{H}_{t+1}^\top\|_2 + \|\mathbf{H}_{t+1} - \mathbf{H}_{t+1}^\top\|_2$ will keep decreasing until it is at most $346\gamma^2$. Now,

$$
\|\mathbf{W}_t\|_2 \leq \|\mathbf{H}_t\|_2 + \|\mathbf{W}^*\|_2 \leq \|\mathbf{H}_{t+1} + \mathbf{H}_{t+1}^\top\|_2 + \|\mathbf{H}_{t+1} - \mathbf{H}_{t+1}^\top\|_2 + \gamma_0 \overset{①}{\leq} (346 + 20)\gamma^2 \leq \gamma
$$

where ① holds as $\gamma_0 = \frac{1}{8000}$. So $\|\mathbf{W}_t\|_2$ is always bounded by $\gamma$.

# I  Proofs for Section D

For notational simplicity, denote

$$
\begin{aligned}
x_j &\triangleq \left(\overline{e_j + w_j} \cdot \overline{e_j + w_j}^\top\right)(w_j^* - w_j), \\
\mathbf{X} &\triangleq (x_1, \cdots, x_d) \tag{19} \\
y_j &\triangleq \left(\mathbf{I} - \overline{e_j + w_j} \cdot \overline{e_j + w_j}^\top\right)(w_j^* - w_j), \\
\mathbf{Y} &\triangleq (y_1, \cdots, y_d) \tag{20} \\
z_j &\triangleq \left(\mathbf{I} - \frac{1}{2}\overline{e_j + w_j} \cdot \overline{e_j + w_j}^\top\right)(w_j^* - w_j), \\
\mathbf{Z} &\triangleq (z_1, \cdots, z_d)
\end{aligned}
$$

We have the following relationship between $x_j, y_j, z_j$.

**Lemma I.1.**

$$
\|z_j\|_2^2 = \frac{1}{4}\|x_j\|_2^2 + \|y_j\|_2^2, \quad \|x_j\|_2^2 + \|y_j\|_2^2 = \|w_j^* - w_j\|_2^2
\tag{21}
$$

*Proof for Lemma I.1.* By definition,

$$
\begin{aligned}
\|z_j\|_2^2 &= \|w_j^* - w_j\|_2^2 \left(\mathbf{I} - \frac{1}{2}\overline{e_j + w_j} \cdot \overline{e_j + w_j}^\top\right)^\top \left(\mathbf{I} - \frac{1}{2}\overline{e_j + w_j} \cdot \overline{e_j + w_j}^\top\right) \\
&= \|w_j^* - w_j\|_2^2 \left(\mathbf{I} - \overline{e_j + w_j} \cdot \overline{e_j + w_j}^\top + \frac{1}{4}\overline{e_j + w_j} \cdot \overline{e_j + w_j}^\top \overline{e_j + w_j} \cdot \overline{e_j + w_j}^\top\right) \\
&= \|w_j^* - w_j\|_2^2 \left(\mathbf{I} - \frac{3}{4}\overline{e_j + w_j} \cdot \overline{e_j + w_j}^\top\right),
\end{aligned}
$$

and similarly

$$\|y_j\|_2^2 = \|w_j^* - w_j\|_2^2 \left(\mathbf{I} - \overline{e_j + w_j} \cdot \overline{e_j + w_j}^\top\right)^\top \left(\mathbf{I} - \overline{e_j + w_j} \cdot \overline{e_j + w_j}^\top\right) = \|w_j^* - w_j\|_2^2 \left(\mathbf{I} - \overline{e_j + w_j} \cdot \overline{e_j + w_j}^\top\right),$$

$$\|x_j\|_2^2 = \|w_j^* - w_j\|_2^2 \left(\overline{e_j + w_j} \cdot \overline{e_j + w_j}^\top\right)^\top \left(\overline{e_j + w_j} \cdot \overline{e_j + w_j}^\top\right) = \|w_j^* - w_j\|_2^2 \left(\overline{e_j + w_j} \cdot \overline{e_j + w_j}^\top\right)$$

The lemma follows. $\qquad\square$

## I.1 Proof for Lemma D.1

In this proof, we heavily use the following trick between the summation of four vector products, and the trace of four matrix products. We give one example below, and other cases are similar.

**Lemma I.2.** $\sum_{i,j} z_j^\top (e_i + w_i^*)(\overline{e_i + w_i^*} - \overline{e_i + w_i})^\top \overline{e_j + w_j} = \mathrm{Tr}\left(\left[\mathbf{Z}^\top(\mathbf{I} + \mathbf{W}^*)\right]\left[(\overline{\mathbf{I} + \mathbf{W}^*} - \overline{\mathbf{I} + \mathbf{W}})^\top \overline{\mathbf{I} + \mathbf{W}}\right]\right).$

*Proof.* By definition, $\mathrm{Tr}(\mathbf{AB}) = \sum_{j=1}^d (\mathbf{AB})_{j,j} = \sum_{i,j} \mathbf{A}_{j,i}\mathbf{B}_{i,j}$. Thus,

$$\mathrm{Tr}\left(\left[\mathbf{Z}^\top(\mathbf{I} + \mathbf{W}^*)\right]\left[(\overline{\mathbf{I} + \mathbf{W}^*} - \overline{\mathbf{I} + \mathbf{W}})^\top \overline{\mathbf{I} + \mathbf{W}}\right]\right) = \sum_{i,j}\left[\mathbf{Z}^\top(\mathbf{I} + \mathbf{W}^*)\right]_{j,i}\left[(\overline{\mathbf{I} + \mathbf{W}^*} - \overline{\mathbf{I} + \mathbf{W}})^\top \overline{\mathbf{I} + \mathbf{W}}\right]_{i,j}$$

By definition, $\left[\mathbf{Z}^\top(\mathbf{I} + \mathbf{W}^*)\right]_{j,i} = z_j^\top(e_i + w_i^*)$, and $\left[(\overline{\mathbf{I} + \mathbf{W}^*} - \overline{\mathbf{I} + \mathbf{W}})^\top \overline{\mathbf{I} + \mathbf{W}}\right]_{i,j} = (\overline{e_i + w_i^*} - \overline{e_i + w_i})^\top \overline{e_j + w_j}$, so the lemma follows. $\qquad\square$

Now we proceed to prove Lemma D.1. We first bound $\sum_{j=1}^d z_j^\top \mathbf{A}_j \overline{e_j + w_j}$ below by splitting $\mathbf{A}_j$ into three parts, and then improve the lower bound in Lemma I.4.

**Lemma I.3.** *If* $\|\mathbf{W}\|_2, \|\mathbf{W}^*\|_2 \le \gamma \le \frac{1}{100}$, *we have*

$$\sum_{j=1}^d z_j^\top \mathbf{A}_j \overline{e_j + w_j} \ge -8\gamma\|\mathbf{W}^* - \mathbf{W}\|_F^2 - \sqrt{\|\mathbf{W}^* - \mathbf{W}\|_f^2 - \frac{3}{4}\|\mathbf{X}\|_F^2}\sqrt{\|\mathbf{W}^* - \mathbf{W}\|_F^2 - \|\mathbf{X}\|_F^2}$$

.

*Proof.* We rewrite $\mathbf{A}_j$ as

$$\mathbf{A}_j = \mathbf{B}_j + \frac{1}{2}\mathbf{C}_j + \mathbf{D}_j \tag{22}$$

where

$$\mathbf{B}_j = \sum_{i \ne j}(e_i + w_i^*)(\overline{e_i + w_i^*} - \overline{e_i + w_i})^\top, \quad \mathbf{C}_j = \sum_{i \ne j}\langle w_i^* - w_i, \overline{e_i + w_i}\rangle \overline{e_i + w_i} \cdot \overline{e_i + w_i}^\top, \quad \mathbf{D}_j = \left(\sum_{i \ne j} z_i \overline{e_i + w_i}^\top\right)$$

For notational simplicity, we also write $\mathbf{B}, \mathbf{C}, \mathbf{D}$ as the corresponding terms with sum $\sum_{i=1}^d$ instead of $\sum_{i \ne j}$, so they do not depend on index $j$. We estimate $\mathbf{B}, \mathbf{C}, \mathbf{D}$ first, then estimate $\mathbf{B}_j, \mathbf{C}_j, \mathbf{D}_j$ respectively by taking the differences.

**1.** From $\mathbf{B}$ to $\mathbf{B}_j$:

$$\sum_{j=1}^d z_j^\top \mathbf{B}\overline{e_j + w_j} = \sum_{i,j} z_j^\top(e_i + w_i^*)(\overline{e_i + w_i^*} - \overline{e_i + w_i})^\top \overline{e_j + w_j}$$

$$\overset{①}{=} \mathrm{Tr}\left(\left[\mathbf{Z}^\top(\mathbf{I} + \mathbf{W})\right]\left[(\overline{\mathbf{I} + \mathbf{W}^*} - \overline{\mathbf{I} + \mathbf{W}})^\top \overline{\mathbf{I} + \mathbf{W}}\right]\right) \overset{②}{\ge} -\left\|(\mathbf{I} + \mathbf{W})^\top \mathbf{Z}\right\|_F \left\|\overline{\mathbf{I} + \mathbf{W}}^\top(\overline{\mathbf{I} + \mathbf{W}^*} - \overline{\mathbf{I} + \mathbf{W}})\right\|_F$$

$$\overset{③}{\ge} -\|\mathbf{I} + \mathbf{W}\|_2 \|\overline{\mathbf{I} + \mathbf{W}}\|_2 \|\mathbf{Z}\|_F \left\|\overline{\mathbf{I} + \mathbf{W}^*} - \overline{\mathbf{I} + \mathbf{W}}\right\|_F \overset{④}{\ge} -\frac{(1 + \gamma)^2}{1 - \gamma}\|\mathbf{Z}\|_F \left\|\overline{\mathbf{I} + \mathbf{W}^*} - \overline{\mathbf{I} + \mathbf{W}}\right\|_F$$

$$\tag{23}$$

where ① uses Lemma I.2, ② uses $\mathrm{Tr}(\mathbf{AB}) \geq -\|\mathbf{A}\|_F \|\mathbf{B}\|_F$, ③ uses $\|\mathbf{AB}\|_F \leq \|\mathbf{A}\|_2 \|\mathbf{B}\|_F$, and ④ uses Lemma F.1. By Lemma E.1 term 1, we have

$$\left\|\overline{\mathbf{I} + \mathbf{W}^*} - \overline{\mathbf{I} + \mathbf{W}}\right\|_F \leq \sqrt{\frac{\sum_{i=1}^d \|y_i\|_2^2}{1 - 2\gamma}} = \frac{\|\mathbf{Y}\|_F}{\sqrt{1 - 2\gamma}} \tag{24}$$

On the other hand,

$$\sum_{j=1}^d z_j^\top (\mathbf{B}_j - \mathbf{B}) \overline{e_j + w_j} = \sum_{j=1}^d z_j^\top (e_j + w_j^*)(\overline{e_j + w_j^*} - \overline{e_j + w_j})^\top \overline{e_j + w_j}$$

$$= \sum_{j=1}^d (w_j^* - w_j)^\top (\mathbf{I} - \frac{1}{2}\overline{e_j + w_j} \cdot \overline{e_j + w_j}^\top)(e_j + w_j^*)(\overline{e_j + w_j^*} - \overline{e_j + w_j})^\top \overline{e_j + w_j}$$

For any vector $x$, $\overline{e_j + w_j} \cdot \overline{e_j + w_j}^\top x$ is the projection of $x$ onto the direction $\overline{e_j + w_j}$, so $\frac{1}{2} \leq \|\mathbf{I} - \frac{1}{2}\overline{e_j + w_j} \cdot \overline{e_j + w_j}^\top\|_2 \leq 1$, and

$$|(w_j^* - w_j)^\top (e_j + w_j^*)(\overline{e_j + w_j^*} - \overline{e_j + w_j})^\top \overline{e_j + w_j}| \overset{\textcircled{1}}{\leq} |(w_j^* - w_j)^\top (e_j + w_j^*)| \frac{\|w_j^* - w_j\|_2^2}{2(1 - 2\gamma)}$$

$$\overset{\textcircled{2}}{\leq} \frac{\|w_j^* - w_j\|_2^3 (1 + \gamma)}{2(1 - 2\gamma)} \leq \frac{\|w_j^* - w_j\|_2^2 (1 + \gamma)\gamma}{1 - 2\gamma} \tag{25}$$

where ① uses Lemma E.1 term 2, and ② uses Cauchy-Schwartz.

Combining (23),(24),(25), we get

$$\sum_{j=1}^d z_j^\top \mathbf{B}_j \overline{e_j + w_j} \geq -\frac{(1 + \gamma)^2}{(1 - \gamma)\sqrt{1 - 2\gamma}} \|\mathbf{Z}\|_F \|\mathbf{Y}\|_F - \frac{(1 + \gamma)\gamma}{1 - 2\gamma} \|\mathbf{W}^* - \mathbf{W}\|_F^2$$

**2.** From $\mathbf{C}$ to $\mathbf{C}_j$:

$$\sum_{j=1}^d z_j^\top \mathbf{C} \overline{e_j + w_j} = \sum_{i,j} z_j^\top \langle w_i^* - w_i, \overline{e_i + w_i} \rangle \overline{e_i + w_i} \cdot \overline{e_i + w_i}^\top \overline{e_j + w_j}$$

$$\overset{\textcircled{1}}{=} \mathrm{Tr}\left(\left[\mathbf{Z}^\top \mathbf{X}\right]\left[\overline{\mathbf{I} + \mathbf{W}}^\top \overline{\mathbf{I} + \mathbf{W}}\right]\right) = \mathrm{Tr}(\mathbf{Z}^\top \mathbf{X}) + \mathrm{Tr}(\mathbf{Z}^\top \mathbf{X}(\overline{\mathbf{I} + \mathbf{W}}^\top \overline{\mathbf{I} + \mathbf{W}} - \mathbf{I}))$$

$$\overset{\textcircled{2}}{\geq} \mathrm{Tr}(\mathbf{Z}^\top \mathbf{X}) - \|\mathbf{Z}\|_F \|\mathbf{X}\|_F \|\overline{\mathbf{I} + \mathbf{W}}^\top \overline{\mathbf{I} + \mathbf{W}} - \mathbf{I}\|_2 \overset{\textcircled{3}}{\geq} \mathrm{Tr}(\mathbf{Z}^\top \mathbf{X}) - \frac{4\gamma}{(1 - \gamma)^2} \|\mathbf{Z}\|_F \|\mathbf{X}\|_F$$

where ① uses Lemma I.2 and $x_j = \langle w_j^* - w_j, \overline{e_j + w_j} \rangle \overline{e_j + w_j}$, ② uses $\mathrm{Tr}(\mathbf{AB}) \geq -\|\mathbf{A}\|_F \|\mathbf{B}\|_F$, and $\|\mathbf{AB}\|_F \leq \|\mathbf{A}\|_2 \|\mathbf{B}\|_F$, and ③ uses Lemma F.1. On the other hand,

$$\sum_{j=1}^d z_j^\top (\mathbf{C} - \mathbf{C}_j) \overline{e_j + w_j} = \sum_{j=1}^d z_j^\top \langle w_j^* - w_j, \overline{e_j + w_j} \rangle \overline{e_j + w_j} \cdot \overline{e_j + w_j}^\top \overline{e_j + w_j}$$

$$= \sum_{j=1}^d z_j^\top \langle w_j^* - w_j, \overline{e_j + w_j} \rangle \overline{e_j + w_j} = \mathrm{Tr}(\mathbf{Z}^\top \mathbf{X})$$

That implies, $\frac{1}{2}\sum_{j=1}^d z_j^\top \mathbf{C}_j \overline{e_j + w_j} \geq -\frac{2\gamma}{(1 - \gamma)^2} \|\mathbf{Z}\|_F \|\mathbf{X}\|_F$.

**3.** From $\mathbf{D}$ to $\mathbf{D}_j$:

$$\sum_{j=1}^d z_j^\top \mathbf{D} \overline{e_j + w_j} = \sum_{i,j} z_j^\top z_i \overline{e_i + w_i}^\top \overline{e_j + w_j} = \mathrm{Tr}\left(\left[\mathbf{Z}^\top \mathbf{Z}\right]\left[\overline{\mathbf{I} + \mathbf{W}}^\top \overline{\mathbf{I} + \mathbf{W}}\right]\right) \geq \frac{(1 - \gamma)^2}{(1 + \gamma)^2} \|\mathbf{Z}\|_F^2$$

where the last inequality holds by Lemma F.1. On the other hand,

$$z_j^\top (\mathbf{D} - \mathbf{D}_j) \overline{e_j + w_j} = \|z_j\|_2^2$$

That gives,

$$\sum_j z_j^\top \mathbf{D}_j \overline{e_j + w_j} \geq -\frac{4\gamma}{(1+\gamma)^2} \|\mathbf{Z}\|_F^2$$

Now, combining $\mathbf{B}_j, \mathbf{C}_j, \mathbf{D}_j$ together, using (22), we have

$$\sum_{j=1}^d z_j^\top \mathbf{A}_j \overline{e_j + w_j} \geq -\frac{(1+\gamma)^2}{(1-\gamma)\sqrt{1-2\gamma}} \|\mathbf{Z}\|_F \|\mathbf{Y}\|_F - \frac{(1+\gamma)\gamma}{1-2\gamma} \|\mathbf{W}^* - \mathbf{W}\|_F^2$$

$$- \frac{2\gamma}{(1-\gamma)^2} \|\mathbf{Z}\|_F \|\mathbf{X}\|_F - \frac{4\gamma}{(1+\gamma)^2} \|\mathbf{Z}\|_F^2$$

By definition, we know $\|\mathbf{X}\|_F \leq \|\mathbf{W}^* - \mathbf{W}\|_F, \|\mathbf{Y}\|_F \leq \|\mathbf{W}^* - \mathbf{W}\|_F, \|\mathbf{Z}\|_F \leq \|\mathbf{W}^* - \mathbf{W}\|_F$, and $\gamma \leq \frac{1}{100}$, so

$$-\frac{(1+\gamma)\gamma}{1-2\gamma}\|\mathbf{W}^* - \mathbf{W}\|_F^2 - \frac{2\gamma}{(1-\gamma)^2}\|\mathbf{Z}\|_F\|\mathbf{X}\|_F - \frac{4\gamma}{(1+\gamma)^2}\|\mathbf{Z}\|_F^2 \geq -7\gamma\|\mathbf{W}^* - \mathbf{W}\|_F^2 \quad (26)$$

Moreover,

$$-\left(\frac{(1+\gamma)^2}{(1-\gamma)\sqrt{1-2\gamma}} - 1\right)\|\mathbf{Z}\|_F\|\mathbf{Y}\|_F \geq -0.05\gamma\|\mathbf{W}^* - \mathbf{W}\|_F^2 \quad (27)$$

Thus, those are small order terms. The only term left is $\|\mathbf{Z}\|_F\|\mathbf{Y}\|_F$. By (21), we know

$$\|\mathbf{Z}\|_F\|\mathbf{Y}\|_F \leq \sqrt{\|\mathbf{W}^* - \mathbf{W}\|_F^2 - \frac{3}{4}\|\mathbf{X}\|_F^2}\sqrt{\|\mathbf{W}^* - \mathbf{W}\|_F^2 - \|\mathbf{X}\|_F^2} \quad (28)$$

Combining (26), (27), (28), we get:

$$\sum_{j=1}^d z_j^\top \mathbf{A}_j \overline{e_j + w_j} \geq -8\gamma\|\mathbf{W}^* - \mathbf{W}\|_F^2 - \sqrt{\|\mathbf{W}^* - \mathbf{W}\|_f^2 - \frac{3}{4}\|\mathbf{X}\|_F^2}\sqrt{\|\mathbf{W}^* - \mathbf{W}\|_F^2 - \|\mathbf{X}\|_F^2}$$

$$\square$$

Now it remains to bound $\sqrt{\|\mathbf{W}^* - \mathbf{W}\|_f^2 - \frac{3}{4}\|\mathbf{X}\|_F^2}\sqrt{\|\mathbf{W}^* - \mathbf{W}\|_F^2 - \|\mathbf{X}\|_F^2}$.

**Lemma I.4.**

$$-\sqrt{\|\mathbf{W}^* - \mathbf{W}\|_F^2 - \frac{3}{4}\|\mathbf{X}\|_F^2}\sqrt{\|\mathbf{W}^* - \mathbf{W}\|_F^2 - \|\mathbf{X}\|_F^2} \geq -1.3\|\mathbf{W}^* - \mathbf{W}\|_F^2 + \|\mathbf{W}^* - \mathbf{W}\|_F\|\mathbf{X}\|_F$$

*Proof.* Consider the function $f(x) = \sqrt{y^2 - \frac{3}{4}x^2}\sqrt{y^2 - x^2} + xy$, where $x \in [0, y]$. It suffices to show that $f(x) \leq 1.3y^2$.

Indeed, we know

$$f'(x) = \frac{x(6x^2 - 7y^2)}{2\sqrt{4y^2 - 3x^2}\sqrt{y^2 - x^2}} + y$$

When $x = 0$, $f'(x) = y > 0$, and when $x \to y$, $f'(x) < 0$. We want to find the place where $f'(x) = 0$, which gives the maximum value. Assume $x = \lambda y$, this is equivalent to solve

$$\lambda y(6(\lambda y)^2 - 7y^2) = -2y\sqrt{4y^2 - 3(\lambda y)^2}\sqrt{y^2 - (\lambda y)^2}$$

Cancel all $y$, and we get the solution $x \approx 0.566y$, where $f(x) \approx 1.2845y^2 < 1.3y^2$. $\square$

*Proof of Lemma D.1.* Combining Lemma I.3 and Lemma I.4, we have proved Lemma D.1. $\square$

## I.2 Proof for Lemma D.2

Again, we first consider the full sum, $g = \sum_{i=1}^{d}(\|e_i + w_i^*\|_2 - \|e_i + w_i\|_2)$.

By Lemma F.3, we have

$$|g - g_j| = |\|e_j + w_j^*\|_2 - \|e_j + w_j\|_2| \leq \|w_j^* - w_j\|_2$$

Thus by Cauchy Schwartz,

$$|(g - g_j)\langle w_j^* - w_j, \overline{e_j + w_j}\rangle| \leq \|w_j^* - w_j\|_2 \|x_j\|_2$$

Summing over $j$, we get

$$\sum_{j=1}^{d}|(g - g_j)\langle w_j^* - w_j, \overline{e_j + w_j}\rangle| \leq \sum_{j=1}^{d}\|w_j^* - w_j\|_2\|x_j\|_2 \leq \|\mathbf{W}^* - \mathbf{W}\|_F\|\mathbf{X}\|_F \qquad (29)$$

where the last inequality is by Cauchy Schwartz.

Now

$$g\sum_{j=1}^{d}\langle w_j^* - w_j, \overline{e_j + w_j}\rangle = g\sum_{j=1}^{d}\langle e_j + w_j^* - e_j + w_j, \overline{e_j + w_j}\rangle$$

$$= g\sum_{j=1}^{d}(\|e_j + w_j^*\|_2 - \|e_j + w_j\|_2 + \langle e_j + w_j^*, \overline{e_j + w_j} - \overline{e_j + w_j^*}\rangle) = g^2 + gb \geq gb \qquad (30)$$

where $b$ is defined to be $\sum_{j=1}^{d}\langle e_j + w_j^*, \overline{e_j + w_j} - \overline{e_j + w_j^*}\rangle$. By Lemma E.1 term 2 we know

$$-\frac{(1+\gamma)\|\mathbf{W}^* - \mathbf{W}\|_F^2}{2(1-2\gamma)} \leq b \leq 0$$

Combining (29), (30), the lemma follows.

$$\sum_{j=1}^{d}\langle g_j\overline{e_j + w_j}, w_j^* - w_j\rangle = \sum_{j=1}^{d}\langle (g_j - g)\overline{e_j + w_j}, w_j^* - w_j\rangle + \sum_{j=1}^{d}\langle g\overline{e_j + w_j}, w_j^* - w_j\rangle$$

$$\geq -\|\mathbf{W}^* - \mathbf{W}\|_F\|\mathbf{X}\|_F + g^2 + gb \geq -\|\mathbf{W}^* - \mathbf{W}\|_F\|\mathbf{X}\|_F - \frac{(1+\gamma)g\|\mathbf{W}^* - \mathbf{W}\|_F^2}{2(1-2\gamma)}$$

## I.3 Proof for Lemma D.3

$$\sum_{j=1}^{d}\langle \mathbf{P}_{3,j}, w_j^* - w_j\rangle = \sum_{j=1}^{d}\langle \frac{\pi}{2}(w_j^* - w_j) - \theta_{j^*,j}(e_j + w_j^*) + \|e_j + w_j^*\|\sin\theta_{j^*,j}\overline{e_j + w_j}, w_j^* - w_j\rangle$$

$$\overset{\textcircled{1}}{=} \sum_{j=1}^{d}\langle \frac{\pi}{2}(w_j^* - w_j) - \theta_{j^*,j}\|e_j + w_j^*\|_2(\overline{e_j + w_j^*} - \overline{e_j + w_j}) + \frac{\alpha_{j^*,j}|\theta_{j^*,j}|^3\|e_j + w_j^*\|\overline{e_j + w_j}}{3}, w_j^* - w_j\rangle$$

$$\overset{\textcircled{2}}{\geq} \frac{\pi}{2}\|\mathbf{W}^* - \mathbf{W}\|_F^2 - \sum_{j=1}^{d}1.001(1+\gamma)\|w_j^* - w_j\|_2^2\|\overline{e_j + w_j^*} - \overline{e_j + w_j}\|_2 - \sum_{j=1}^{d}0.335(1+\gamma)\|w_j^* - w_j\|_2^4$$

$$\overset{\textcircled{3}}{\geq} \frac{\pi}{2}\|\mathbf{W}^* - \mathbf{W}\|_F^2 - \sum_{j=1}^{d}\frac{1.001(1+\gamma)}{\sqrt{1-2\gamma}}\|w_j^* - w_j\|_2^3 - \sum_{j=1}^{d}0.335(1+\gamma)\|w_j^* - w_j\|_2^4$$

$$\overset{\textcircled{4}}{\geq} \left(\frac{\pi}{2} - 0.021\right)\|\mathbf{W}^* - \mathbf{W}\|_F^2$$

where ① uses Taylor's Theorem for $\sin\theta_{j^*,j}$, so we know $|\alpha_{j^*,j}| \leq 1$. ② uses Lemma E.1 term 3 and Cauchy Schwartz, ③ uses Lemma E.1 term 1, ④ holds since $\gamma \leq \frac{1}{100}$, and the two small order terms can be bounded by $0.021\|\mathbf{W}^* - \mathbf{W}\|_F^2$.

# J Proofs for Section 2

## J.1 Proof for Lemma 2.5

By the updating rule, we have

$$
\mathbb{E}\|\mathbf{W}_{t+1} - \mathbf{W}^*\|_F^2 = \mathbb{E}\|\mathbf{W}_t - \mathbf{W}^* - \eta\mathbf{G}_t\|_F^2 = \mathbb{E}\|\mathbf{W}_t - \mathbf{W}^*\|_F^2 - 2\langle\mathbf{W}_t - \mathbf{W}^*, \eta\nabla f(\mathbf{W})\rangle + \eta^2\|\mathbf{G}_t\|_F^2
$$
$$
\leq \mathbb{E}\|\mathbf{W}_t - \mathbf{W}^*\|_F^2 - 2\langle\mathbf{W}_t - \mathbf{W}^*, \eta\nabla f(\mathbf{W})\rangle + \eta_t^2 G^2 \leq (1 - 2\eta\delta)\mathbb{E}\|\mathbf{W}_t - \mathbf{W}^*\|_F^2 + \eta^2 G^2
$$

Now if $\eta\delta\mathbb{E}\|\mathbf{W}_t - \mathbf{W}^*\|_F^2 \geq \eta^2 G^2$, we know the $\mathbb{E}\|\mathbf{W}_t - \mathbf{W}^*\|_F^2$ will decrease by a factor of $(1 - \eta\delta)$ for every step. Otherwise, although it could increase, we know

$$
\mathbb{E}\|\mathbf{W}_t - \mathbf{W}^*\|_F^2 \leq \frac{\eta G^2}{\delta}
$$

By setting $\eta = \frac{(1+\alpha)\log T}{\delta T}$, we know after $T$ steps, either $\mathbb{E}\|\mathbf{W}_T - \mathbf{W}^*\|_F^2$ is already smaller than $\frac{\eta G^2}{\delta} = \frac{(1+\alpha)\log TG^2}{\delta^2 T}$, or it is decreasing by factor of $(1 - \eta\delta)$ for every step, which means

$$
\mathbb{E}\|\mathbf{W}_T - \mathbf{W}^*\|_F^2 \leq \mathbb{E}\|\mathbf{W}_0 - \mathbf{W}^*\|_F^2 (1-\eta\delta)^T \leq D^2 e^{-\eta\delta T} = D^2 e^{-(1+\alpha)\log T} = \frac{D^2 T^{-\alpha}}{T} \leq \frac{(1+\alpha)\log TG^2}{\delta^2 T}.
$$

The last inequality holds since

$$
T^\alpha \log T \geq \frac{D^2\delta^2}{(1+\alpha)G^2}
$$

Thus, $\mathbb{E}\|\mathbf{W}_T - \mathbf{W}^*\|_F^2$ will be smaller than $\frac{(1+\alpha)\log TG^2}{\delta^2 T}$.