[Reviews · NeurIPS 2017]

Reviewer 1



This paper provides convergence guarantees for two-layer residual networks with ReLU activations using SGD. This is an important problem and will be interesting to NIPS community. In particular, authors show that when the input is Gaussian and the ground truth parameters (the residual part W^*) are small enough, SGD will converge to the global optimal, which is also the ground truth in the considered setting. Preliminary experiments verify the claim on both synthetic data and real-world data. The paper is in general well written and a nice proof flowchart is provided to help readers. The two-phase analysis is interesting and novel, where in the first phase a potential function is constructed and proved to decrease with the iterations of SGD and in the second phase a classical SGD analysis is employed following one-point strong convexity. Potential issue in proof: We found a potential flaw in the proof for Lemma 2.5 (the convergence analysis of SGD). In line 778 of the appendix, the second equality should lead to $\langle W_t - W^*, \eta G_t \rangle$ instead of $\langle W_t - W^*, \eta \nabla f(W) \rangle$. These two formulations are equivalent to each other if they are in expectation, but the current results in Lemma 2.5 are not in expectation. Comparison to result of [ZSJ+2017]: a) it's important to note that the works are independent as that paper has not been presented so far, b) however, both the papers provide very similar Guarantees as this paper require "small" norm of the residual parameter while [ZSJ+2017] require good enough initialization, c) this paper however provides analysis of SGD which is more practical algorithm than GD. Overall, the paper presents an interesting result for two-layer neural networks and should be of interest to the NIPS community. However, the proof might have an issue that requires clarification. Refs: [ZSJ+ 2017] Recovery Guarantees for One-hidden-layer Neural Networks, ICML 2017, https://arxiv.org/pdf/1706.03175.pdf

Reviewer 2



This paper discusses the convergence property of SGD for two layer NN with ReLU and identity mapping. The authors argue that SGD converges to a global minimizer from (almost) arbitrary initialization in two phases: * Phase 1: W is drifting in wrong directions but a potential function g(W) = || W* + I ||_{2, 1} - || W* + I ||_{2, 1} decreases * Phase 2: W converges to W* at a sublinear rate The analysis is a combination of local one-point restricted strong convexity (used to show Phase 2) + guarantees for entering this benign region (Phase 1). I didn't check the proofs line by line, but I'm a bit unclear about the statements of Lemma 5, I would kindly ask the authors to elaborate the following: First, W^t needs to satisfy the one-point strongly convex property for all t. We need to control ||W^t||_2 < gamma -- the upper bound for ||W^*||_2. I think for this you need some smoothness assumption. Could you please point me where how do you ensure this? Second, the authors assumes that the iterates W^t stays in the one-point RSS region with diameter D. In line 174, you mentioned D = sqrt(d)/50, where d is the input dimension. This means the number of iterations you need must be larger than dimension d. I think this might be a bit weak, as many convergence results for nonconvex optimization are dimension free. Thanks!

Reviewer 3



This paper proves the convergence of a stochastic gradient descent algorithm from a suitable starting point to the global minimizer of a nonconvex energy representing the loss of a two-layer feedforward network with rectified linear unit activation. In particular, the algorithm is shown to converge in two phases, where phase 1 drives the iterates into a one-point convex region which subsequently leads to the actual convergence in phase 2. The findings, the analysis, and particularly the methodology for proving the convergence (in 2 phases) are very interesting and definitely deserve to be published. The entire proof is extremely long (including a flowchart of 15 Lemmas/Theorems that finally allow to show the main theorem in 25 pages of proofs), and I have to admit that I did not check this part. I have some questions on the paper and suggestions to further improve the manuscript: - Figure 1 points out the different structure of the considered networks, and even the abstract already refers to the special structure of "identity mappings". However, optimizing for W (vanilla network) is equivalent to optimizing for (W+I), such that the difference seems to lie in different starting points only. If this is correct, I strongly suggest to emphasize the starting point rather than the identity mapping. In this case, please shorten the connection to ResNet. - There is a fundamental difference between the architecture shown in figure 1 (which is analyzed in the proofs), and the experiments in sections 5.1, 5.4, and 5.5. In particular, the ResNet trick of an identity shortcut behaves fundamentally different if the identity is used after a nonlinearity (e.g. BatchNorm in Figure 6 and ReLu in the ResNet Paper), opposed to an equivalent reformulation of the original network in Figure 1. Moreover, sections 5.1, 5.4, and 5.5 consider deep networks that have not been analyzed theoretically. Since your theoretical results deal with the 2-layer architecture shown in Figure 1, I suggest to reduce the numerical experiments that go well beyond the theoretically studied case, and instead explain the latter in more detail. It certainly contains enough material. In particular, it would be interesting to understand how close real-world data is to satisfying the assumptions of Theorem 3.1. - For a paper of the technical precision such as the one presented here, I think one should not talk about taking the derivative of a ReLU function without a comment. The ReLU is not differentiable whenever any component is zero - so what exactly is \nabla L(W)? In a convex setting one would consider a subgradient, but what is the right object here? A weak derivative? - Related to the previous question, subgradient descent on a convex function already requires the step size to go to zero in order to state convergence results. Can you comment on why you can get away with constant step sizes? - The plots in Figures 7 and 8 are difficult to read and would benefit from more detailed explanations, and labeled axes. - I recommend to carefully check the manuscript with respect to third person singular verbs and plural nouns. Also, some letters are accidently capitalized after a comma.